# BalancedDPO: Adaptive Multi-Metric Alignment

**Dipesh Tamboli**[†][*]   **Souradip Chakraborty**[‡]   **Aditya Malusare**[†]   **Biplab Banerjee**[§]
**Amrit Singh Bedi**[¶]   **Vaneet Aggarwal**[†]

[†] **Purdue University**   [‡] **University of Maryland**
[§] **Indian Institute of Technology Bombay**   [¶] **University of Central Florida**

Reviewed on OpenReview: `https://openreview.net/forum?id=8HRID5VLQw`

## Abstract

Diffusion models have achieved remarkable progress in text-to-image generation, yet aligning them with human preference remains challenging due to the presence of multiple, sometimes conflicting, evaluation metrics (e.g., semantic consistency, aesthetics, and human preference scores). Existing alignment methods typically optimize for a single metric or rely on scalarized reward aggregation, which can bias the model toward specific evaluation criteria. To address this challenge, we propose BALANCEDDPO, a framework that achieves multi-metric preference alignment within the Direct Preference Optimization (DPO) paradigm. Unlike prior DPO variants that rely on a single metric, BALANCEDDPO introduces a majority-vote consensus over multiple preference scorers and integrates it directly into the DPO training loop with dynamic reference model updates. This consensus-based formulation avoids reward-scale conflicts and ensures more stable gradient directions across heterogeneous metrics. Experiments on Pick-a-Pic, PartiPrompt, and HPD datasets demonstrate that BALANCED-DPO consistently improves preference win rates over the baselines across Stable Diffusion 1.5, Stable Diffusion 2.1 and SDXL backbones. Comprehensive ablations further validate the benefits of majority-vote aggregation and dynamic reference updating, highlighting the method's robustness and generalizability across diverse alignment settings.

## 1 Introduction

Text-to-image (T2I) diffusion models achieved remarkable progress in generating images that exhibit strong semantic alignment, aesthetic appeal, and overall visual quality (Ramesh et al., 2022; Saharia et al., 2022; Rombach et al., 2022; Nichol et al., 2021). However, aligning T2I models with diverse and nuanced human preferences remains an open challenge, particularly in real-world scenarios where multiple, often conflicting, objectives must be optimized simultaneously. For example, consider the prompt: "*A warning sign for a flooded street.*" A successful model must accurately depict the key safety element—a clear, legible warning sign—while also ensuring the context, such as a visibly flooded street, is realistic and unambiguous. Balancing clarity, semantic alignment, and aesthetic quality is crucial in such safety-critical scenarios, where misaligned or ambiguous outputs could lead to misinterpretation and potential harm.

Existing alignment methods, such as DiffusionDPO (Wallace et al., 2024), leverage direct preference optimization (DPO) (Rafailov et al., 2024) to fine-tune T2I diffusion models using labeled preferences.

Although the methods discussed above are effective for individual metrics, these approaches struggle to balance competing objectives such as aesthetics, text-image alignment, and human preference scores. A naive solution is to perform multi-objective optimization (Zhou et al., 2024), which involves directly modifying the loss function to combine multiple metrics or rewards. However, this approach faces critical challenges: **(C1)** *reward rescaling issues*, as rewards often operate on different scales (aesthetics and HPS scores have different ranges), leading to dominance by specific rewards/metrics and biased alignment; **(C2)** *conflicting*

---

*Code: `https://github.com/Dipeshtamboli/BalancedDPO`

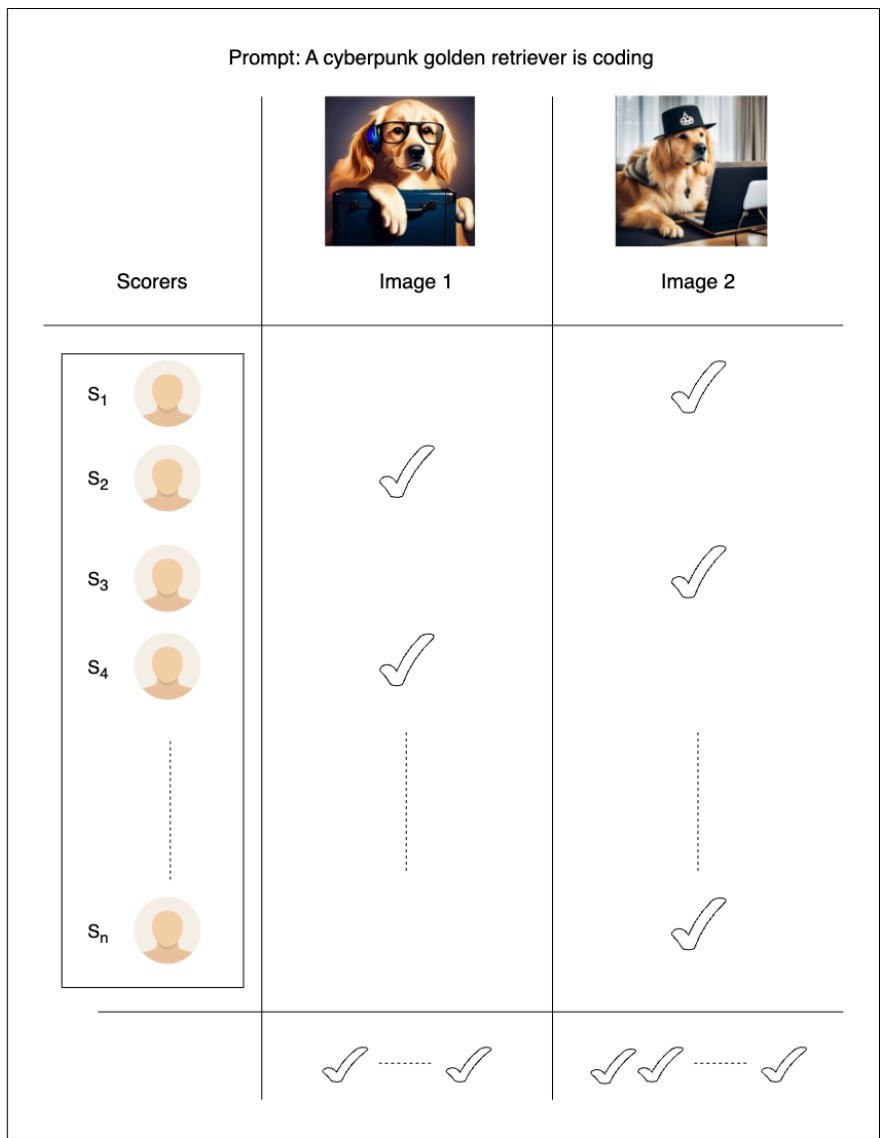

Figure 1: Overview of the BALANCEDDPO framework for the prompt "A cyberpunk golden retriever is coding." Multiple reward models (Scorers $S_1$ to $S_n$) evaluate a pair of generated images. Each scorer casts a binary vote for their preferred image. Image 2, receiving the majority of votes, is designated the **Winner**. This majority-voting mechanism resolves the scaling disparities between metrics (e.g., $S_1 \in [0, 1]$ vs. $S_2 \in [1, 100]$) and avoids gradient conflicts, outperforming traditional normalization strategies as demonstrated in our results.

*gradients*, where improving one reward may compromise others, making convergence unstable; and **(C3)** *pipeline complexity*, as integrating multiple metrics would requires substantial modifications to the standard DPO framework, complicating implementation and reducing model generalization.

To address these limitations, we propose BALANCEDDPO, a novel multi-metric alignment framework that extends the existing DPO (Rafailov et al., 2024) methodology with minimal changes. The core innovation lies in defining how preferences are aggregated; instead of mixing multiple rewards directly in the loss function, which results in scaling and dominance issues, as shown in our ablation studies, BALANCEDDPO operates in the preference distribution space. By leveraging a majority-voting-based aggregation strategy inspired by social choice theory (Prasad, 2018), our method curates a balanced preference dataset that

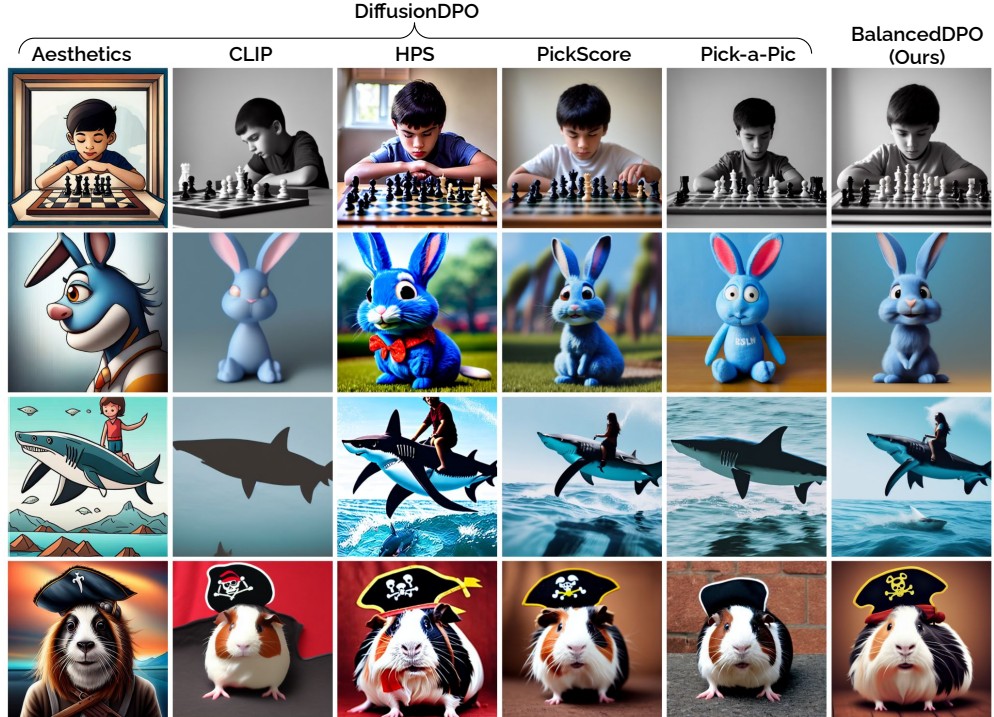

Figure 2: Comparison of images generated by models trained on image-text pairs from the Pick-a-Pic dataset and preference labels based on different score metrics (Aesthetics (Schuhmann, 2022), CLIP (Ramesh et al., 2022), HPS (Wu et al., 2023), PickScore (Kirstain et al., 2023), Pick-a-Pic labels (Kirstain et al., 2023)), and BALANCEDDPO (combining all metrics) across four prompts:*"a boy playing chess"*, *"A Pixar style blue rabbit"*, *"person riding a shark"*, and *"pirate guinea pig"*. Results show that single-metric models often fail in either aesthetics or prompt alignment. For instance, the Aesthetics model generates a cartoonish "shark rider," while Pick-a-Pic labels produce a realistic but incomplete image. In contrast, BALANCEDDPO achieves superior performance across all cases, highlighting the benefit of multi-metric optimization.

dynamically combines diverse metrics such as CLIP score, aesthetic quality, and Human Preference Score (HPS). As shown in Fig. 1, multiple scorers are used to cast votes in favor of which image they find superior. This relative choice mitigates the need for normalization over widely varying scoring scales and helps us continuously incorporate feedback into the alignment process. This approach captures the nuanced trade-offs between multiple objectives, allowing the model to align with diverse preferences without requiring complex modifications to the DPO pipeline. Fig 2 demonstrates how single-metric models can fail in alignment or aesthetics, while BALANCEDDPO achieves multi-metric optimization through the aggregation strategy.

We summarize our contributions as follows:

- **A novel paradigm for multi-metric alignment:** We introduce BALANCEDDPO, a framework that performs preference-based alignment in text-to-image diffusion models using a multi-metric majority-vote consensus. Unlike prior approaches that optimize a single reward or scalarized combination, our method aggregates diverse scoring models (e.g., CLIP, HPS, PickScore, Aesthetic) into a unified binary preference label used directly within the DPO loss (Eq. 7–9). This design avoids scale-dependent gradient conflicts and enables robust alignment across heterogeneous objectives.

- **Dynamic reference model updating:** In contrast to traditional T2I alignment methods that fix a base reference model, BALANCEDDPO periodically updates the reference model with the current model parameters (Alg. 1), maintaining stability while allowing exploration in new alignment

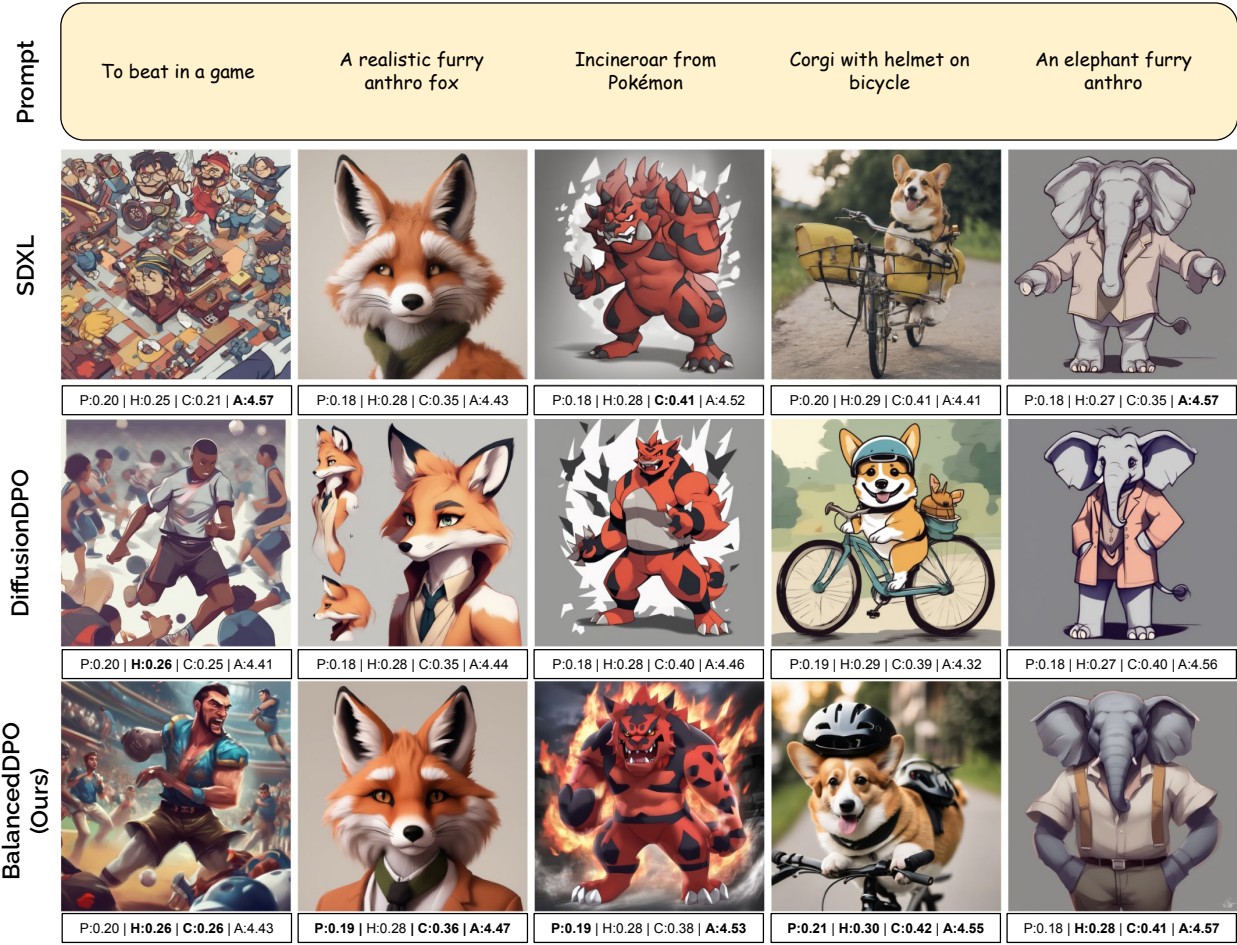

Figure 3: **Pick-a-Pic (Kirstain et al., 2023) comparison for SDXL.** Comparison of images generated by various SDXL fine-tunes (specify versions if applicable) and BALANCEDDPO (Ours) on the Pick-a-Pic dataset across diverse prompts. BALANCEDDPO generally creates images that are more realistic and possess finer details. They are also superior in terms of prompt alignment and visual attractiveness. The columns emphasize BALANCEDDPO's strength in areas like background details, object details, prompt adherence, and overall aesthetic. The values displayed underneath each image indicate PickScore (P), Human Preference Score (H), CLIP score (C), and Aesthetic score (A).

directions. As demonstrated in our experiments, this mechanism substantially improves convergence and consistency across multiple datasets and evaluation metrics.

- **Empirical Evaluations:** We conduct extensive experiments demonstrating that BALANCEDDPO outperforms prior alignment methods across both Stable Diffusion 1.5, SD 2.1, and SDXL backbones. For instance, on the Pick-a-Pic dataset (Kirstain et al., 2023), BALANCEDDPO achieves an 85.2% win rate in HPS, a 78.4% win rate in CLIP score, and a 69.4% win rate in aesthetic quality, outperforming Diffusion-DPO by 14.5%, 20.2%, and 16.6%, respectively. On the newly added SD 2.1 experiments, BALANCEDDPO further demonstrates consistent gains: it achieves 62.33 in HPS, 65.33 in CLIP, 69.33 in PickScore, and 65.33 in Aesthetic score when compared against SD 2.1, and surpasses DiffusionDPO by 53.67, 65.67, 69.67, and 70.00 in the respective metrics. Consistent improvements are also observed on the PartiPrompt dataset (Yu et al., 2022), as discussed in 4.2.

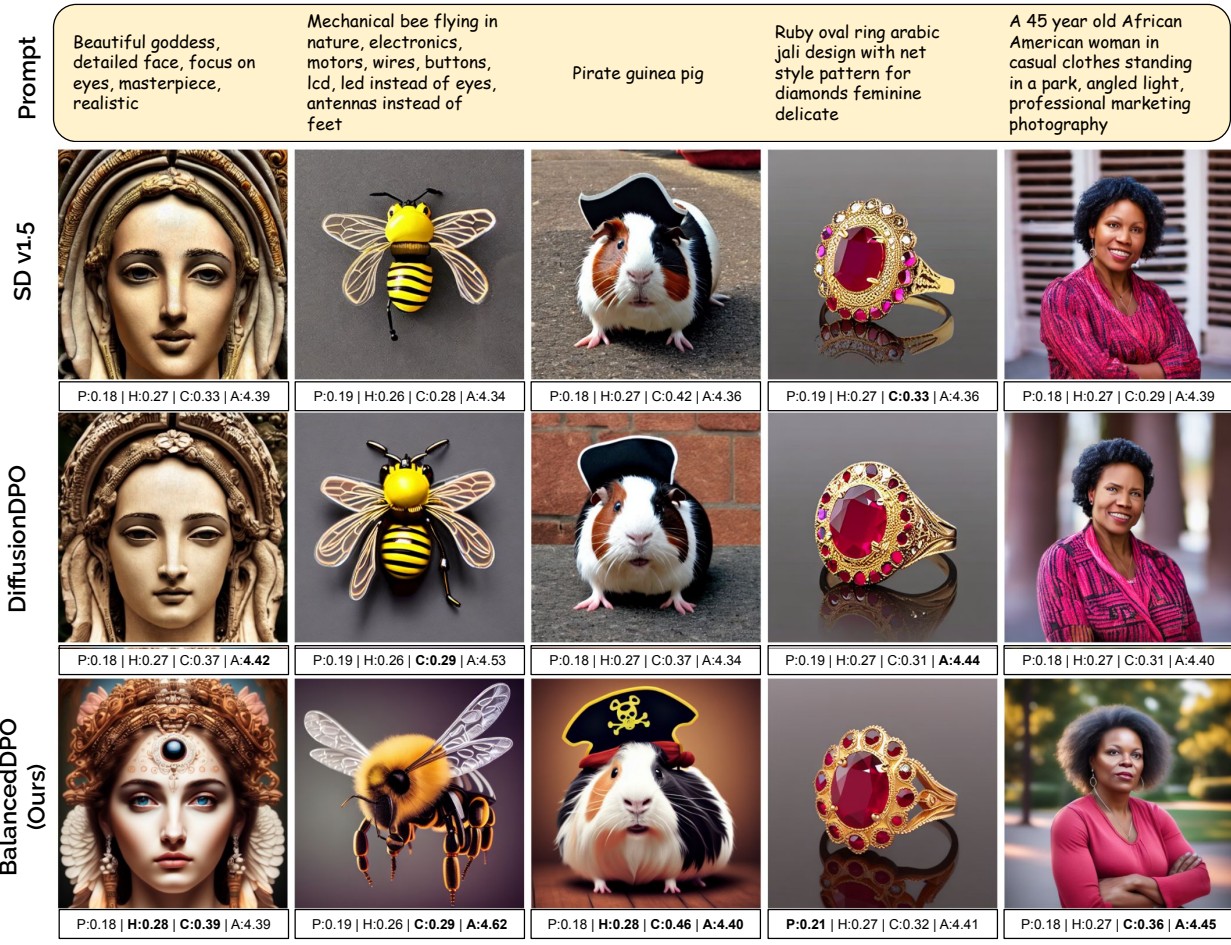

Figure 4: **Pick-a-Pic (Kirstain et al., 2023) comparison.** Comparison of images generated by SD1.5, DiffusionDPO, and BALANCEDDPO (Ours) across various prompts. BALANCEDDPO consistently produces more realistic and detailed outputs, outperforming the other models in aligning with prompts and visual appeal. Each column highlights BALANCEDDPO's superior performance in aspects like facial detail, dynamic motion, adherence to prompt details, and image reflection. The scores below each image represent PickScore (P), Human Preference Score (H), CLIP score (C), and Aesthetic score (A).

## 2    Related Works

**RLHF Based Alignment of Diffusion Models:** Reinforcement Learning from Human Feedback (RLHF) has been pivotal in aligning diffusion models to human preferences, optimizing output quality beyond standard training. Foundational work in this field (Fan et al., 2024; Wang et al., 2022; Lee et al., 2023) extended RLHF methods to domain-specific diffusion models, ensuring outputs met specialized constraints, such as those in high-resolution photorealistic generation. This improvement was realized in these models through training with Proximal Policy Optimization (PPO) (Schulman et al., 2017), allowing iterative tuning that addressed visual fidelity and semantic alignment (Zheng et al., 2023a). Black et al. (Black et al., 2023) highlighted enhancements in complex generation scenarios by integrating RLHF with training strategies for diffusion models in specific reinforcement learning contexts. Chen et al. (Chen et al., 2024) explored the application of RLHF to diffusion-based text-to-speech synthesis, using mean opinion scores as a proxy loss and introducing a loss-guided reinforcement learning policy optimization of the diffusion model to improve speech quality and naturalness. Furthermore, Dong et al. (Dong et al., 2023) proposed a framework that uses RLHF to quantify

human preferences and guides diffusion planning for the customization of zero-shot behavior, effectively aligning agent behaviors with diverse human preferences. But all of the RLHF-based approaches require access to reward function during fine-tuning as well as restricted to the use of PPO algorithm, which exhibits instabilities in practice (Zheng et al., 2023b).

**Direct Preference Based Alignment:** DPO (Rafailov et al., 2023) is designed to align generative models with human aesthetic and content preferences by directly optimizing the model's parameters based on human feedback, thus eliminating the need for an intermediate reward model. DiffusionDPO (Wallace et al., 2024) extended DPO from a language-model-based formulation (Rafailov et al., 2023) to account for the unique likelihood structures of diffusion models, utilizing the evidence lower bound to derive a differentiable objective. This approach is then applied to fine-tune the Stable Diffusion XL (Podell et al., 2023) model using the Pick-a-Pic (Kirstain et al., 2023) dataset, resulting in significant improvements in visual appeal and prompt alignment as evaluated by human judges. The direct preference for Denoising Diffusion Policy Optimization (D3PO) method (Yang et al., 2023), which fine-tunes diffusion models directly based on human feedback without the need for a reward model, demonstrates effectiveness in reducing image distortions and generating safer images. Other work extends this idea to relative (Gu et al., 2024) and diffusion step-aware (Liang et al., 2024) preference optimization, as well as to other modalities like structure-based drug design (Cheng et al., 2024). But existing DPO-based approaches for diffusion models are limited to single preference feedback, and aligned models suffer from suboptimal performance with respect to multiple metrics, which are important in real-world applications.

## 3 Problem Formulation

### 3.1 Background and Preliminaries

**DiffusionDPO:** DPO (Rafailov et al., 2024) is a method originally developed for aligning language models with human preferences (Rafailov et al., 2024). DPO focuses on learning from pairs of ranked outputs, where one sample is preferred over another. Let $x_0^w$ and $x_0^l$ represent the "winning" and "losing" samples, respectively, for a given conditioning $c$. The Bradley-Terry (BT) model (Bradley & Terry, 1952) is then used to express human preferences as

$$p_{\text{BT}}(x_0^w \succ x_0^l | c) = \sigma(r(c, x_0^w) - r(c, x_0^l)), \tag{1}$$

where $\sigma$ is the sigmoid function, and $r(c, x_0)$ represents the latent reward model that scores the preference of sample $x_0$ given conditioning $c$. Reward model is learned via maximum likelihood estimation for binary classification:

$$L_{\text{BT}}(r) = -\mathbb{E}_{c, x_0^w, x_0^l} \left[ \log \sigma(r(c, x_0^w) - r(c, x_0^l)) \right], \tag{2}$$

where the expectation is taken over text prompts $c$ and data pairs $(x_0^w, x_0^l)$ from a dataset annotated with human preferences. The direct preference optimization in (Rafailov et al., 2023) bypasses the reward learning step and proposes to directly optimize the model from preference feedback. DiffusionDPO (Wallace et al., 2024) adapted DPO specifically for diffusion models, introducing a preference optimization objective that aligns the reverse process $p_\theta(x_{0:L})$ with the human-preferred distribution. The optimization objective Eq. (12) of (Wallace et al., 2024) is expressed as:

$$\mathcal{L}_{\text{Diff-DPO}}(\theta) = -\frac{2}{\beta} \mathbb{E}_1 \log \sigma \left( \mathbb{E}_2 \left[ \beta \log \frac{p_\theta(x_0^w)}{p_{\text{ref}}(x_0^w)} - \beta \log \frac{p_\theta(x_0^l)}{p_{\text{ref}}(x_0^l)} \right] \right), \tag{3}$$

where the outer expectation $\mathbb{E}_1$ is with respect to $(x_0^w, x_0^l) \sim \mathcal{D}$, inner expectation $\mathbb{E}_2$ is with respect to $x_{1:L}^w \sim p_{\theta(x_{1:T}^w | x_0^w)}, x_{1:L}^l \sim p_{\theta(x_{1:T}^l | x_0^l)}$ where $x_{1:L}$ denotes the latents of the diffusion path leading to $x_0$, $\sigma(\cdot)$ denotes the sigmoid function, $\beta \geq 0$ is a regularization constant, and $\mathcal{D}$ represents the dataset of paired human preferences.

**Remark:** The objective in equation 3 aligns the model based on a given preference dataset, which typically reflects a single utility metric. For example, the dataset might prioritize selecting images with better aesthetic

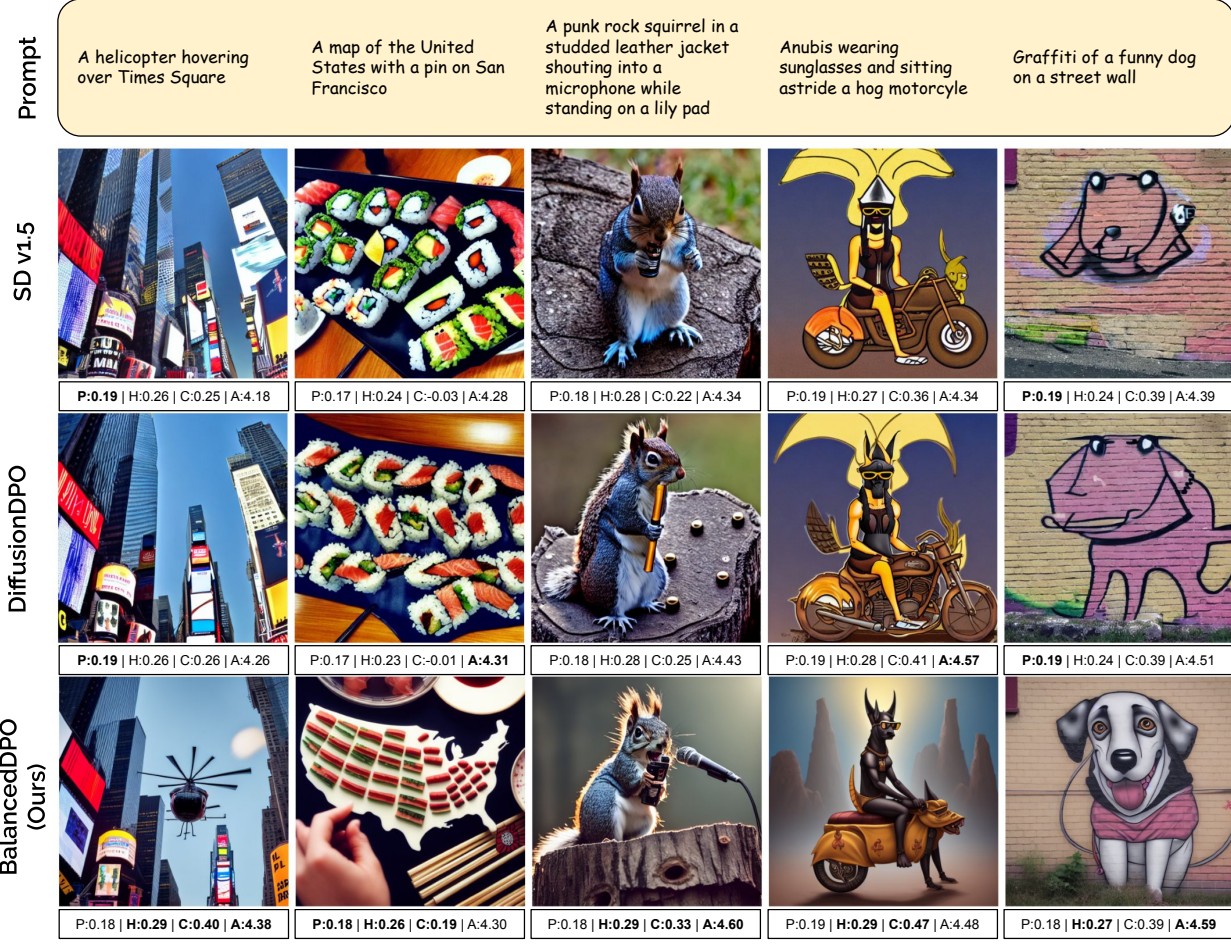

Figure 5: **PartiPrompt (Yu et al., 2022) comparison.** Comparison of images generated by SD1.5, DiffusionDPO, and BALANCEDDPO (Ours) on out-of-distribution prompts from the PartiPrompt dataset. BALANCEDDPO consistently generates more accurate and realistic outputs, including specific elements like a helicopter, microphone, and lifelike dog, while the other models produce incomplete or irrelevant results.

quality over others. However, this approach fails to capture the multifaceted requirements of real-world image generation, where preferences often span multiple attributes such as semantic alignment with text prompts, aesthetic appeal, diversity of generated content, and adherence to human preference scores. Balancing these diverse and sometimes conflicting objectives is essential for creating robust and versatile text-to-image models suitable for practical applications.

## 3.2 Multi-metric Alignment for Diffusion Models

We start by highlighting the limitations of single-utility-based preference optimization methods in the context of the text-image diffusion models (see Figure 2). For a given text-to-image diffusion model $p_\theta(x_0|c)$ where $x_0$ is the generated image and $c$ conditioning text prompt, the goal is to align the model outputs (which is a generated image) to human preferences and diverse attributes. As discussed earlier, traditional approaches typically optimize a single reward function $r(c, x_0)$, which reflects a specific aspect of user preference, such as semantic alignment, aesthetic quality (Wallace et al., 2024). The standard single-reward based alignment can

be expressed as:

$$\max_{p_\theta} \mathbb{E}_{c \sim \mathcal{D}_c, x_0 \sim p_\theta(x_0|c)}[r(c, x_0)] - \beta D_{\mathrm{KL}}[p_\theta(x_0|c) \| p_{\mathrm{ref}}(x_0|c)], \tag{4}$$

where $\mathcal{D}_c$ represents the distribution over prompts and $r(c, x_0)$ is the reward function capturing alignment with a particular preference criterion. For diffusion models, since reward is evaluated for the final generated image out of trajectory $x_{o:T}$, we can define the reward function as $r(c, x_0) = \mathbb{E}_{x_{1:T}}[R(c, x_{o:T})]$, where $R$ is the reward for the full chain $x_{o:T}$ as mentioned in (Wallace et al., 2024). For the sake of discussion and to motivate the contributions in this section, it is sufficient to consider the reward model $r(c, x_0)$ only.

To this end, we remark that the alignment formulation in equation 4 is limited in capturing multi-faceted human preferences because optimizing solely for $r(c, x_0)$ may result in overfitting to specific attributes, reducing the general performance. As a naive (vanilla) solution to it, we define the multi-objective preference optimization problem next.

**Multi-objective Preference Optimization**: We assume access to diverse reward functions $\{r_k(c, x_0)\}_{k=1}^K$ each representing a distinct alignment metric. These metrics may include human preference score (HPS), CLIP score, aesthetic quality, etc. In an ideal scenario, there exists a latent, unknown aggregation function $f$ that combines these reward functions to reflect the true composite preference which can be captured as

$$s(c, x_0) = f(r_1(c, x_0), r_2(c, x_0), \cdots, r_K(c, x_0)), \tag{5}$$

where $s(c, x_0)$ is the aggregated score representing the true overall alignment quality of the generated image $x_0$ given the prompt $c$. Unfortunately, it is almost impossible to have access to this function $f(r_1, r_2 \cdots, r_k)$.

**A fundamental challenge:** Due to the unavailability of the true aggregation function $f(r_1, r_2 \cdots, r_k)$ directly applying DPO methods becomes challenging since we cannot compute a single, invertible reward (refer to (Rafailov et al., 2023; Wallace et al., 2024) for the invertibility requirement) that accurately captures the latent preferences, leading to issues in defining a consistent optimization objective, similar to one we have in equation 3. On the other hand, direct aggregation approaches, such as a weighted sum or averaging of reward functions would face several issues: **1. Unknown weights:** The optimal weights for combining rewards are typically unknown, and estimating these weights requires a separate optimization process, which can introduce additional loss terms and complexity. **2. Scale and Dominance:** Each reward function $r_k$ may operate on different scales, which can cause a direct summation to disproportionately favor certain metrics leading to sub-optimal and biased alignment. **3. Conflicting Objectives:** Directly optimizing the aggregated function can introduce conflicts, where improvement in one reward, complicates convergence.

Therefore, an effective and robust preference aggregation mechanism is required to mitigate the issues which is agnostic to the scale and dominance of individual reward attributes.

### 3.3 Our Method: Majority Voting Aggregation

**Key idea:** To mitigate the issues discussed above, we take motivation from Social choice theory majority rule (Prasad, 2018) to perform aggregation of the preference labels after collecting them from different rewards for the same pairs of data points. This approach is agnostic to the reward scale and the need for optimal weight parameters required in direct aggregation methods.

**Our proposed method**: We have $K$ attributes given as $r_1, r_2 \cdots r_K$ and a fixed dataset of prompts and image-pairs as $\mathcal{D} = \{(c, x_0^1, x_0^2)\}$. First, we define a score function $s_k(c, x_0^1, x_0^2)$ corresponding to each reward model as

$$s_k(c, x_0^1, x_0^2) = \begin{cases} 1 & \text{if } r_k(c, x_0^1) \geq r_k(c, x_0^2) \text{ by} \\ -1 & \text{otherwise.} \end{cases} \tag{6}$$

Based on the above scores, we define our aggregation strategy motivated by the majority rule as

$$s(c, x_0^1, x_0^2) = \mathrm{sign}\left(\sum_{i=1}^K s_k\right), \tag{7}$$

which indicates that $s = 1$, when majority of the attributes prefers $x_0^1$ over $x_0^2$ and $s = -1$, otherwise. This aggregates these individual preferences via majority voting. We utilize the score $s(c, x_0^1, x_0^2)$ to decide whether the pair $(x_0^1, x_0^2)$ is $(x_0^w, x_0^l)$ (when $s(c, x_0^1, x_0^2) = 1$) or $(x_0^l, x_0^w)$ (for $s(c, x_0^1, x_0^2) = -1$). We do it for all the data points in data set $\mathcal{D}$. Using score function definition in equation 7 and the DiffusionDPO loss in equation 3, it holds that

$$\mathcal{L}_{\text{agg}}(\theta) = -\mathbb{E}_{(c, x_0^1, x_0^2) \sim \mathcal{D}} \left[ \log \sigma \left( \beta \cdot s \cdot [l_\theta(c, x_0^1, x_0^2)] \right) \right], \tag{8}$$

where, $l_\theta(c, x_0^1, x_0^2) = \mathbb{E}_2 \left[ \log \frac{p_\theta(x_0^1|c)}{p_\theta(x_0^1|c)} - \log \frac{p_\theta(x_0^2|c)}{p_\theta(x_0^2|c)} \right]$ for notation simplicity and $\mathbb{E}_2$ is same as in equation 3. Next, with this definition, the gradient of the loss function is given by

$$\nabla_\theta \mathcal{L}_{\text{agg}}(\theta) = -\mathbb{E}_{\mathcal{D}} \left[ \beta \cdot s \cdot \sigma \left( -\beta \cdot l_\theta(c, x_0^1, x_0^2) \right) \nabla_\theta l_\theta(c, x_0^1, x_0^2) \right]. \tag{9}$$

The above formulation results in a simplified and efficient objective almost similar to direct preference optimization and agnostic to the scale of the reward functions.

**Advantage over direct aggregation strategies:** We show that using a naive direct aggregation strategy (see Table 3) might lead to an issue in aggregating gradient due to potentially conflicting objectives leading to slower convergence. The naive direct aggregation strategy given as optimizing the direct preference loss

$$\mathcal{L}_{\text{da}}(\theta) = -\mathbb{E}_{(c, x_0^1, x_0^2) \sim \mathcal{D}} \left[ \sum_k w_k \log \sigma \left( \beta \cdot s_k \cdot l_\theta(c, c, x_0^1, x_0^2) \right) \right], \tag{10}$$

where $w_k, s_k$ represents the weights (unknown) and preference for the $k^{\text{th}}$ attribute. Similarly, the gradient of the objective boils down to

$$\nabla_\theta \mathcal{L}_{\text{da}}(\theta) = -\mathbb{E}_{\mathcal{D}} \left[ \sum_k w_k \beta s_k \sigma \left( -\beta l_\theta(c, x_0^1, x_0^2) \right) \nabla_\theta l_\theta(c, x_0^1, x_0^2) \right]. \tag{11}$$

To mathematically illustrate the gradient contradiction, consider a scenario with two reward models, $r_i$ and $r_j$, that disagree on a preference ($s_i = 1, s_j = -1$). In the naive aggregation strategy (Eq. 11), the resulting gradient term for a given pair is proportional to $(w_i \cdot s_i + w_j \cdot s_j) \nabla_\theta l_\theta$. If the weights $w_i, w_j$ are similar, the terms $w_i(1)$ and $w_j(-1)$ effectively cancel each other out, leading to a vanishingly small update. This "gradient cancellation" stalls the optimization process, as the model receives no clear direction despite the presence of preference data.

In contrast, our majority-vote signal $s = \text{sign}(\sum s_k)$ in Eq. 7 resolves this disagreement in the preference space. It distills the conflicting rewards into a single, high-confidence directional indicator, ensuring that $\nabla_\theta \mathcal{L}_{\text{agg}}$ provides a non-vanishing update toward the majority-preferred manifold. This prevents the destructive interference of gradients and leads to the more efficient convergence observed in our experiments.

**Updating the based reference model:** Another important aspect of our approach is that since we are already catering to multiple diverse rewards through the aggregation strategy, it makes sense to allow our model to move a little away from the reference model base, and hence we also replace $p_{\text{ref}}$ with the recent $p_{\theta_t}$ during training and it provides remarkable performance, as shown in the experimental section next.

**Theoretical Rationale:** Before performing the gradient analysis (Eqs. 10-11), we highlight the underlying motivation for our approach. This analysis shows that naive score-weighting can lead to conflicting gradients, whereas majority voting ensures consistent update direction across metrics.

**The Rationale for Binary Discretization:** While converting real-valued rewards into binary votes ($s_k \in \{-1, 1\}$) results in a loss of magnitude information, it provides a critical advantage in multi-metric alignment: *directional consensus*. In practice, reward models like HPS, Aesthetic Score, and CLIP operate on fundamentally different distributions and scales. Traditional scalar aggregation (Eq. 5) is highly sensitive to these heterogeneous ranges, often requiring exhaustive hyperparameter tuning of weights $w_k$ to prevent a single high-variance metric from dominating the gradient (the Reward Rescaling issue, C1). By discretizing the

---

**Algorithm 1** Majority Voting Aggregation for BALANCEDDPO

---

**Require:**
    Initial model parameters $\theta_0$
    Dataset $\mathcal{D} = \{(c, x_0^1, x_0^2)\}$
    Reward functions $\{r_k\}_{k=1}^K$
    Learning rate $\eta$, temperature $\beta$
    Reference model update interval $T$
**Ensure:** Optimized parameters $\theta^*$
  1: Initialize reference model $p_{\text{ref}} \leftarrow p_{\theta_0}$
  2: **for** training step $t = 1$ to $N_{\text{steps}}$ **do**
  3:      Sample batch $\{(c, x_0^1, x_0^2)\} \sim \mathcal{D}$
  4:      **for** each $(c, x_0^1, x_0^2)$ in batch **do**
  5:          Compute attribute preferences:
  6:          **for** $k = 1$ to $K$ **do**
  7:              $s_k \leftarrow \begin{cases} 1 & \text{if } r_k(x_0^1, c) \geq r_k(x_0^2, c) \\ -1 & \text{otherwise} \end{cases}$
  8:          **end for**
  9:          Aggregate preferences: $s \leftarrow \text{sign}\left(\sum_{k=1}^K s_k\right)$
10:      **end for**
11:      Compute loss gradient: $\nabla_\theta \mathcal{L}_{\text{agg}} \leftarrow -\mathbb{E}\left[\beta s \sigma(-\beta l_\theta) \nabla_\theta l_\theta\right]$
12:      Update parameters: $\theta_{t+1} \leftarrow \theta_t - \eta \nabla_\theta \mathcal{L}_{\text{agg}}$
13:      **if** $t \equiv 0 \mod T$ **then**
14:          Update reference model: $p_{\text{ref}} \leftarrow p_{\theta_t}$
15:      **end if**
16: **end for**

---

preference into a vote, we achieve scale-invariance. This ensures that the model optimizes for the preference direction agreed upon by the majority, making the training process robust to outliers and preventing "reward hacking" where a model might exploit the numerical peaks of a single noisy metric at the expense of others.

**Flexibility of the Voting Schema:** While we employ a simple majority vote in our experiments, the BALANCEDDPO framework is inherently flexible. In scenarios where specific reward models are deemed more reliable or critical than others, our method naturally supports *weighted voting*. By redefining the consensus signal as $s = \text{sign}(\sum_{k=1}^K w_k s_k)$, where $w_k$ represents the confidence or priority of the $k$-th metric, practitioners can incorporate domain-specific priors into the alignment process without reverting to the instabilities of scalar-sum optimization.

### 3.4 BalancedDPO Aggregation

The proposed method implements the majority voting strategy through three key phases, as detailed in Algorithm 1:

1. **Preference Collection** (Lines 3-8): For each image pair $(x_0^1, x_0^2)$ and prompt $c$, we query all $K$ attribute-specific reward models to obtain binary preferences $s_k \in \{-1, 1\}$. This converts continuous reward values into directional indicators, eliminating scale sensitivity.

2. **Majority Aggregation** (Line 9): The function $s = \text{sign}(\sum s_k)$ implements Condorcet's Majority Principle by selecting the option favored in pairwise comparisons and satisfies Independence of Irrelevant Alternatives (IIA) by ensuring decisions remain unaffected by the presence of other options, making it robust to outliers.

3. **Gradient Update** (Lines 11-13): The unified preference signal $s$ drives a single gradient update through the BALANCEDDPO loss, avoiding conflicting updates from multiple rewards. The reference model refresh (Line 14) follows curriculum learning principles, gradually tightening the KL constraint as the model improves.

Notably, the algorithm maintains $\mathcal{O}(K)$ complexity for preference collection but only $\mathcal{O}(1)$ complexity for gradient updates, making it as efficient as single-attribute DPO training. The periodic reference model update (Line 14) prevents premature convergence while maintaining training stability.

This consensus-based formulation provides a principled theoretical foundation for resolving the primary challenges of multi-metric alignment. By distilling multiple objectives into a single, unified preference signal $s$, BALANCEDDPO ensures that the diffusion model converges toward image distributions most consistently favored across a diverse panel of "voters" (reward models). Furthermore, the relative preference between image pairs remains stable and decoupled from the presence of other samples in the batch. This theoretical framework directly addresses the *Conflicting Gradients* (C2) issue; rather than navigating a landscape of divergent updates from heterogeneous metrics, the model follows a singular, high-confidence gradient direction rooted in social choice consensus.

## 4 Experiments and Analysis

### 4.1 Experiment Setup

**Datasets**  We use the Pick-a-Pic dataset (Kirstain et al., 2023) for training, which provides pairwise preference images for each caption. Preference labels are derived from an ensemble of score models (PickScore (Kirstain et al., 2023), HPS (Wu et al., 2023), CLIP (Ramesh et al., 2022), and Aesthetic (Schuhmann, 2022)), creating a multi-faceted image quality evaluation. For evaluation, we use Pick-a-Pic, PartiPrompt (Yu et al., 2022), and HPD (Wu et al., 2023). Safe prompts are extracted using the Huggingface (Li, 2023) model. Detailed dataset information is provided in Appendix A.

**Model**  We validate BALANCEDDPO using Stable Diffusion v1.5 (SD1.5) (Rombach et al., 2022) and compare it with the publicly available SD1.5 weights from the Hugging Face model hub (Rombach et al.).

**Hyperparameters**  Our model is trained for 2000 steps with updates to $p_{\text{ref}}$ every 100 steps, using the AdamW optimizer (Loshchilov, 2017) on a single NVIDIA A100 GPU. The batch size is effectively 128, and the learning rate is set to $1 \times 10^{-8}$ with warmup, following the DiffusionDPO (Wallace et al., 2024) setup with adjustments for efficiency.

**Evaluation**  We evaluate BALANCEDDPO against SD1.5 and DiffusionDPO by generating five images per caption using random seeds. Images are assessed using four metrics: Human Preference Score (HPS), CLIP score, PickScore, and Aesthetic score. The highest score for each metric and prompt is selected to mitigate variations. Win rates are computed for model comparisons (BALANCEDDPO vs. SD1.5, BALANCEDDPO vs. DiffusionDPO, DiffusionDPO vs. SD1.5), representing the proportion of times one model's best score surpasses the other's. Detailed evaluation procedures are available in the supplemental material (see Appendix D).

### 4.2 Results

Table 1: Comparison of win rates (%) for **DiffusionDPO vs SD1.5**, **BalancedDPO vs SD1.5**, and **BalancedDPO vs DiffusionDPO** across three datasets (**Pick-a-Pic**, **PartiPrompt**, **HPD**) and four evaluation metrics (HPS, CLIP score, PickScore, Aesthetic score). BALANCEDDPO outperforms SD1.5 and DiffusionDPO in most cases, with significant improvements in CLIP and Aesthetic scores on Pick-a-Pic and PartiPrompt. In HPD, while DiffusionDPO leads in HPS, BALANCEDDPO excels in CLIP, PickScore, and Aesthetic scores.

| Model | Pick-a-Pic | | | | PartiPrompt | | | | HPD | | | |
|---|---|---|---|---|---|---|---|---|---|---|---|---|
| | HPS | CLIP | PickScore | Aesthetic | HPS | CLIP | PickScore | Aesthetic | HPS | CLIP | PickScore | Aesthetic |
| DiffusionDPO vs SD1.5 | 70.71 | 57.24 | 56.23 | 52.86 | 67.76 | 56.24 | 50.75 | 50.49 | **74.00** | 50.67 | 47.33 | 56.00 |
| BALANCEDDPO vs SD1.5 | **85.19** | **78.45** | **64.31** | **69.36** | **70.06** | **65.19** | **56.33** | **62.18** | 72.67 | **72.67** | **60.67** | **63.33** |
| BALANCEDDPO vs DiffusionDPO | 73.06 | 73.40 | 65.99 | 69.70 | 61.20 | 62.80 | 56.95 | 62.62 | 57.33 | 69.33 | 64.00 | 60.00 |

Table 2: Comparison of win rates (%) for **DiffusionDPO vs SDXL**, **BalancedDPO vs SDXL**, and **BalancedDPO vs DiffusionDPO** across three datasets (**Pick-a-Pic**, **PartiPrompt**, **HPD**) and four evaluation metrics (HPS, CLIP score, PickScore, Aesthetic score). BALANCEDDPO outperforms SDXL and DPO in most cases, with significant improvements in CLIP and Aesthetic scores on Pick-a-Pic and PartiPrompt. In HPD, while DPO leads in HPS, BALANCEDDPO excels in CLIP, PickScore, and Aesthetic scores.

| Model | Pick-a-Pic | | | | PartiPrompt | | | | HPD | | | |
|---|---|---|---|---|---|---|---|---|---|---|---|---|
| | HPS | CLIP | PickScore | Aesthetic | HPS | CLIP | PickScore | Aesthetic | HPS | CLIP | PickScore | Aesthetic |
| DiffusionDPO vs SDXL | 78.00 | 59.00 | 45.00 | 21.00 | 76.00 | 58.00 | 52.00 | 26.00 | 78.00 | 55.00 | 38.00 | 14.00 |
| BALANCEDDPO vs SDXL | **98.00** | **88.00** | **62.00** | **44.00** | **96.00** | **94.00** | **76.00** | **66.00** | **97.00** | **78.00** | **50.00** | **42.00** |
| BALANCEDDPO vs DiffusionDPO | 82.00 | 77.00 | 64.00 | 81.00 | 88.00 | 86.00 | 80.00 | 93.00 | 80.00 | 74.00 | 75.00 | 83.00 |

Table 3: Ablation study on the Pick-a-Pic dataset comparing BALANCEDDPO with models trained on individual metrics (HPS, CLIP, PickScore, Aesthetic) and Vanilla Aggregation. The table shows win rates against SD1.5 and DiffusionDPO across four metrics. Models trained on a single metric (e.g., HPS) excel in that metric but perform poorly in others. BALANCEDDPO, using a multimetric consensus approach, consistently outperforms all ablated versions, highlighting the advantage of optimizing across multiple metrics for superior overall performance.

| Label Preference | Win against SD1.5 | | | | Win against DiffusionDPO | | | |
|---|---|---|---|---|---|---|---|---|
| | HPS | CLIP | PickScore | Aesthetic | HPS | CLIP | PickScore | Aesthetic |
| HPS | **89.90** | 49.49 | 56.90 | 14.48 | **87.54** | 44.78 | 58.25 | 12.46 |
| CLIP | 11.45 | 57.91 | 58.92 | 63.30 | 5.05 | 51.52 | 55.56 | 67.00 |
| PickScore | 80.47 | 58.25 | 43.77 | 47.81 | 73.74 | 52.19 | 44.11 | 43.77 |
| Aesthetic | 52.19 | 26.94 | 56.23 | 57.58 | 39.06 | 20.54 | 55.22 | 56.23 |
| Ours (Random Score Function) | 83.83 | 56.22 | 49.15 | 38.04 | 78.45 | 45.11 | 45.79 | 35.01 |
| Ours (Vanilla/Naive Aggregation) | 69.70 | 31.65 | 53.54 | 53.54 | 54.21 | 26.60 | 53.20 | 52.19 |
| Ours (Normalized Aggregation) | 84.28 | 42.54 | 60.29 | 62.23 | 59.87 | 60.49 | 61.38 | 63.47 |
| Ours (BALANCEDDPO) | 85.19 | **78.45** | **64.31** | **69.36** | 73.06 | **73.40** | 65.99 | **69.70** |

**Comparison with baseline** Table 1 compares the win rates of BALANCEDDPO (Ours) and DiffusionDPO across three datasets: Pick-a-Pic, PartiPrompt, and HPD. In the Pick-a-Pic dataset, BALANCEDDPO significantly outperforms DiffusionDPO with a win rate of 85.19% in HPS (compared to 70.71%) and 78.45% in CLIP (compared to 57.24%). Similarly, in the PartiPrompt dataset, BALANCEDDPO achieves higher scores across all metrics, particularly in HPS (70.06% vs. 67.76%) and Aesthetic (62.18% vs. 50.49%). In the HPD dataset, while DiffusionDPO has a slight edge in HPS (74.00% vs. 72.67%), BALANCEDDPO outperforms it in CLIP (72.67% vs. 50.67%), PickScore (60.67% vs. 47.33%), and Aesthetic (63.33% vs. 56.00%). Overall, BALANCEDDPO consistently surpasses DiffusionDPO across most metrics and datasets, demonstrating superior alignment with human preferences and improved visual quality.

Table 2 further presents pairwise relative preference scores comparing DiffusionDPO vs. SDXL, BalancedDPO vs. SDXL, and BalancedDPO vs. DiffusionDPO across three datasets (Pick-a-Pic, PartiPrompt, HPD) and four evaluation metrics (HPS, CLIP, PickScore, and Aesthetic). BalancedDPO consistently achieves higher preference relative to both SDXL and DiffusionDPO across nearly all metrics and datasets. Specifically,

Table 4: Ablation study on the Pick-a-pic dataset comparing BALANCEDDPO with its ablated versions and DiffDPO across four metrics: Human Preference Score (HPS), CLIP score, PickScore, and Aesthetic score. Results show that removing either the multimetric scoring system or reference distribution update significantly lowers performance, especially in Aesthetic and CLIP scores. Both ablated versions still outperform DiffDPO, with the full BALANCEDDPO model achieving the highest scores across all metrics, highlighting the importance of both components.

| Method | $p_{ref}$ update | Multi-Metric | HPS | CLIP | PickScore | Aesthetic |
|---|---|---|---|---|---|---|
| DiffusionDPO | ✗ | ✗ | 70.71 | 57.24 | 56.23 | 52.86 |
| BALANCEDDPO | ✗ | ✓ | 62.96 | 60.94 | 47.14 | 53.87 |
| | ✓ | ✗ | 82.49 | 62.29 | 46.13 | 43.43 |
| | ✓ | ✓ | **85.19** | **78.45** | **64.31** | **69.36** |

in Pick-a-Pic, BalancedDPO is strongly preferred over SDXL (98% HPS, 88% CLIP, 62% PickScore, 44% Aesthetic) and also clearly preferred relative to DiffusionDPO (82% HPS, 77% CLIP, 64% PickScore, 81% Aesthetic). On PartiPrompt, the relative preference remains high, favoring BalancedDPO significantly over SDXL (96% HPS, 94% CLIP, 76% PickScore, 66% Aesthetic) and DiffusionDPO (88% HPS, 86% CLIP, 80% PickScore, 93% Aesthetic). Similarly, for HPD, BalancedDPO demonstrates a strong relative preference over SDXL (97% HPS, 78% CLIP, 50% PickScore, 42% Aesthetic) and over DiffusionDPO (80% HPS, 74% CLIP, 75% PickScore, 83% Aesthetic). DiffusionDPO itself is consistently preferred relative to SDXL, though generally by smaller margins compared to BalancedDPO.

To further assess backbone generalization, we extend our evaluation to the Stable Diffusion 2.1 model (see supplementary Section C.1). As summarized in Table 5, BalancedDPO again exhibits substantial gains over both SD 2.1 and DiffusionDPO across all metrics on the Pick-a-Pic dataset, achieving 62.33 in HPS, 65.33 in CLIP, 69.33 in PickScore, and 65.33 in Aesthetic. Compared to DiffusionDPO, which records 57.33, 47.67, 50.67, and 45.67 respectively, these results confirm that the proposed multimetric consensus and dynamic reference updating remain effective even on more recent diffusion backbones.

**Pick-a-Pic dataset analysis** Figure 4 compares the outputs of SD1.5, DiffusionDPO, and BALANCEDDPO (Ours) across various prompts, highlighting the qualitative differences between the models. In the first row (*"Beautiful goddess, detailed face..."*), BALANCEDDPO generates a more realistic and lifelike face, while SD1.5 and DiffusionDPO produce stone-like, statue-like faces. Similarly, in the second row (*"Mechanical bee flying..."*), BALANCEDDPO accurately captures the action of a bee in flight, unlike SD v1.5 and DiffusionDPO, where the bees remain static. In the third row (*"Pirate guinea pig"*), only BALANCEDDPO successfully includes the pirate cap as specified in the prompt, while the other models miss this key detail. For the fourth row (*"Ruby oval ring..."*), BALANCEDDPO produces a clearer and more complete reflection of the ring compared to SD1.5 and DiffusionDPO. Lastly, in the fifth row (*"A 45-year-old African American woman..."*), BALANCEDDPO preserves facial structure and details better than the other models, which produce less realistic faces. Overall, BALANCEDDPO demonstrates superior alignment with prompts and visual quality across all categories, often outperforming SD v1.5 and DiffusionDPO in both aesthetic appeal and prompt adherence.

**PartiPrompt dataset analysis:** Figure 5 compares the outputs of SD1.5, DiffusionDPO, and BALANCED-DPO (Ours) across five prompts from the PartiPrompt dataset, which is out-of-distribution (OOD) for these models. In the first row, with the prompt *"A helicopter hovering over Times Square"*, only BALANCEDDPO successfully generates a helicopter, while both SD1.5 and DiffusionDPO fail to include it. In the second row, for the prompt *"A map of the United States with a pin on San Francisco"*, all models generate a recognizable map, but none correctly place the pin. However, BALANCEDDPO produces a more detailed and visually appealing map compared to the others. In the third row, with the prompt *"A punk rock squirrel in a studded leather jacket shouting into a microphone while standing on a lily pad"*, BALANCEDDPO accurately includes the microphone as specified in the prompt, while SD1.5 and DiffusionDPO generate irrelevant objects like sushi or random items. In the fourth row, for the prompt *"Anubis wearing sunglasses and sitting astride a hog motorcycle"*, BALANCEDDPO produces a more realistic Anubis with aviator sunglasses, while SD1.5 and DiffusionDPO generate less detailed, cartoonish versions. Finally, in the fifth row, for the prompt *"Graffiti of a funny dog on a street wall"*, BALANCEDDPO generates a more lifelike dog in graffiti style compared to the abstract representations from SD1.5 and DiffusionDPO. Overall, BALANCEDDPO consistently outperforms SD1.5 and DiffusionDPO in both prompt alignment and visual quality, showcasing superior aesthetic appeal and adherence to textual descriptions across all prompts.

**HPD dataset analysis:** Figure 6 illustrates the qualitative performance of SD1.5, DiffusionDPO, and BALANCEDDPO (Ours) across five prompts from the HPD (Wu et al., 2023) dataset. The results highlight the limitations of SD1.5 and DiffusionDPO in generating images that are both semantically aligned with the prompts and aesthetically appealing. For example, in the first row ("A blue airplane in a blue, cloudless sky"), BALANCEDDPO generates a more realistic airplane with accurate proportions and lighting compared to the less convincing outputs from SD v1.5 and DiffusionDPO.

In the second row ("A gummy chameleon hanging on a tree branch"), BALANCEDDPO captures both the gummy texture and natural lighting effectively, while the outputs from other models fail to balance realism

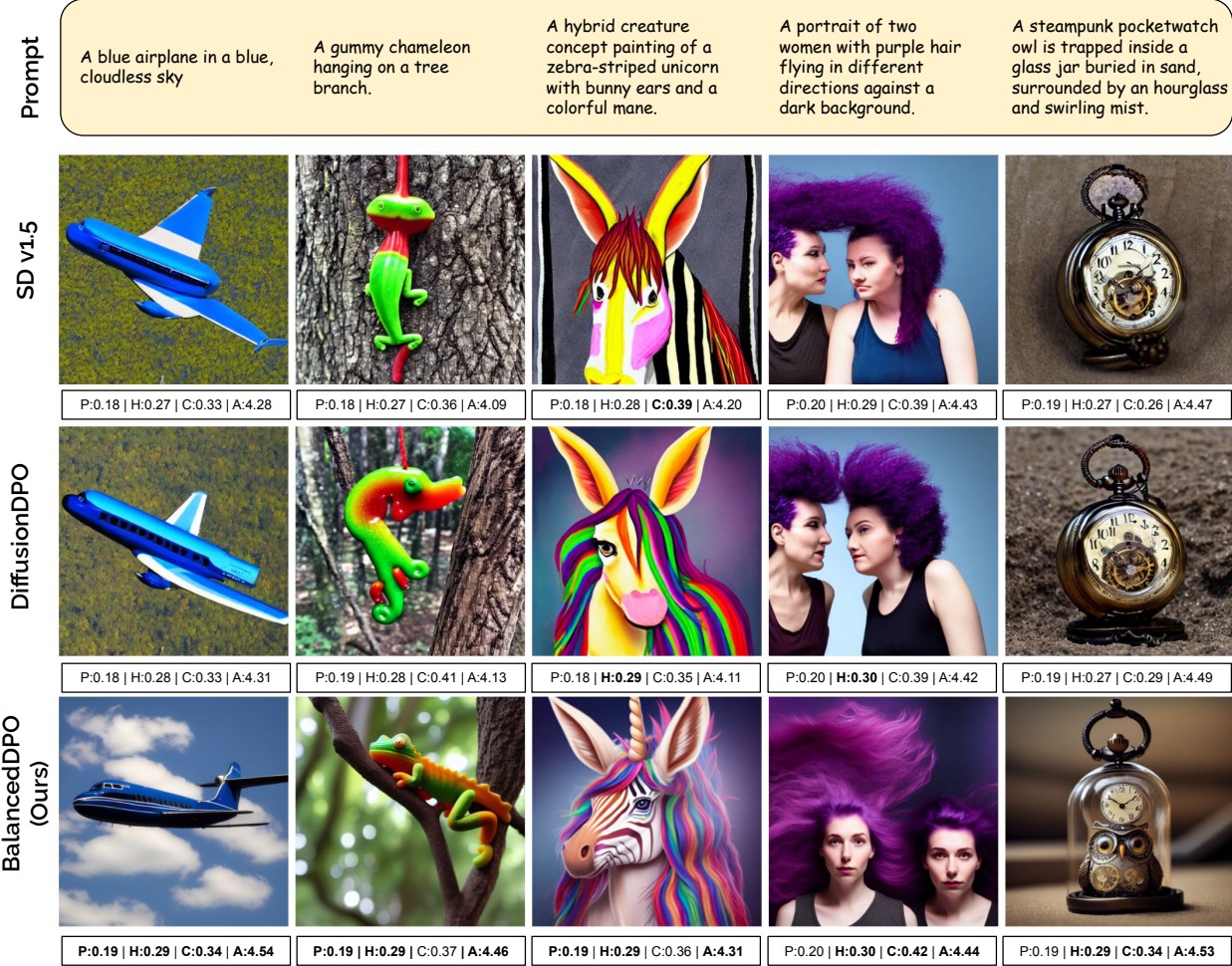

Figure 6: Comparison of images generated by SD v1.5, DiffusionDPO, and BALANCEDDPO (Ours) on prompts from the HPD dataset. Each row corresponds to a specific prompt, and the columns display the generated outputs for each method. BALANCEDDPO consistently produces more realistic, detailed, and aesthetically pleasing images while aligning better with the given prompts compared to SD v1.5 and DiffusionDPO. The scores below each image represent PickScore (P), Human Preference Score (H), CLIP score (C), and Aesthetic score (A).

and prompt adherence. Similarly, in the third row ("A hybrid creature concept painting of a zebra-striped unicorn with bunny ears and a colorful mane"), BALANCEDDPO produces a vibrant and visually appealing image, whereas other models generate less pleasing outputs.

The fourth row ("A portrait of two women with purple hair flying in different directions against a dark background") demonstrates BALANCEDDPO's ability to preserve facial details while adhering to prompt specifics, outperforming SD1.5 and DiffusionDPO, which produces less realistic faces with poor text alignment (background is not dark). Finally, in the fifth row ("A steampunk pocket watch owl is trapped inside a glass jar buried in sand, surrounded by an hourglass and swirling mist"), BALANCEDDPO generates an intricate and visually compelling image that captures both steampunk aesthetics and prompt details, whereas other models fail to achieve this level of detail.

Overall, BALANCEDDPO consistently outperforms SD1.5 and DiffusionDPO across all prompts by producing images that are better aligned with textual descriptions while maintaining superior visual quality across all evaluated metrics.

**Effectiveness of different score methods:**

To rigorously evaluate the necessity of majority voting, we compare BALANCEDDPO against three alternative aggregation strategies in Table 3:

- **Random Score Function:** At each training iteration, one of the $K$ reward functions is randomly sampled to determine the preference $s$, representing a stochastic approach to multi-metric alignment.

- **Vanilla Aggregation:** The preference is determined by the sign of the unweighted sum of raw scalar rewards: $s = \text{sign}(\sum_{k=1}^{K} r_k)$. This baseline is highly susceptible to the "Reward Rescaling" (C1) issue.

- **Normalized Aggregation:** To mitigate scale dominance, rewards are Z-score normalized based on batch statistics ($z_k = \frac{r_k - \mu_k}{\sigma_k}$) before being summed to determine the preference signal.

Table 3 highlights the effectiveness of training models using individual score functions (HPS, CLIP, PickScore, Aesthetic) versus combining all score functions as done in BALANCEDDPO (Ours). Models trained on HPS score show strong performance in their respective metric but underperform in others. HPS-trained model achieves high HPS scores (89.90% against SD1.5 and 87.54% against DiffusionDPO) but performs poorly in Aesthetic (14.48% against SD1.5). Similarly, the CLIP-trained model excels in CLIP score (57.91% against SD1.5) but struggles with HPS. In contrast, BALANCEDDPO, which optimizes across all metrics simultaneously, consistently outperforms across most metrics, achieving the highest scores in CLIP (78.45%), PickScore (64.31%), and Aesthetic (69.36%) against SD1.5, and similarly outperforming DiffusionDPO across all metrics.

Even when utilizing all four scoring methods, Vanilla Aggregation and Normalized Aggregation do not yield optimal results. The Random Score Function approach, which involves selecting a random score function at each iteration, shows improved performance over individual metrics but still falls short of BALANCEDDPO's performance. Notably, Vanilla Aggregation struggles to balance the different metrics effectively, with lower win rates across the board compared to BALANCEDDPO. Normalized Aggregation, while showing some improvement over Vanilla Aggregation, still fails to match the consistent high performance of BALANCEDDPO across all metrics.

This demonstrates that combining multiple score functions in a meaningful way (BALANCEDDPO) leads to better overall performance and balanced results across different evaluation criteria.

**Effectiveness of Balanced Multimetric Alignment and Reference Model update:** Table 4 highlights the importance of Multimetric Alignment and frequent updates to the reference model ($p_{\text{ref}}$) in achieving superior performance. When removing the multimetric scoring system, the model still performs well in HPS (82.49%) and CLIP (62.29%) as it is able to chase and overfit the human preference score but underperforms in PickScore (46.13%) and Aesthetic (43.43%). Similarly, without frequent updates to $p_{\text{ref}}$, the model achieves a higher Aesthetic score (53.87%) but suffers in HPS (62.96%) and PickScore (47.14%). In contrast, BALANCEDDPO (with both multimetric alignment and frequent $p_{\text{ref}}$ updates) consistently outperforms other variants across all metrics, achieving the highest scores in HPS (85.19%), CLIP (78.45%), PickScore (64.31%), and Aesthetic (69.36%). This demonstrates that optimizing multiple metrics simultaneously and frequently updating the reference model are crucial for achieving balanced and superior performance across different evaluation criteria. These results underscore the effectiveness of our BALANCEDDPO approach in simultaneously optimizing multiple image quality metrics. Unlike DiffusionDPO, which shows uneven improvements across different metrics, BALANCEDDPO consistently outperforms across all evaluated dimensions. This balanced improvement suggests that our method successfully addresses the limitations of previous approaches, achieving a more holistic enhancement of image generation quality. The consistent performance across both the in-distribution Pick-a-Pic dataset and the out-of-distribution PartiPrompt dataset further demonstrates the robustness and generalizability of our BALANCEDDPO method. This strong OOD performance is particularly significant, as it indicates that BALANCEDDPO can effectively handle a wide range of prompt styles and complexities beyond its training distribution.

The synergistic relationship between multimetric alignment and the dynamic reference model ($p_{\text{ref}}$) update is critical. In single-metric DPO, frequent $p_{\text{ref}}$ updates can lead to instability or "reward hacking" as the

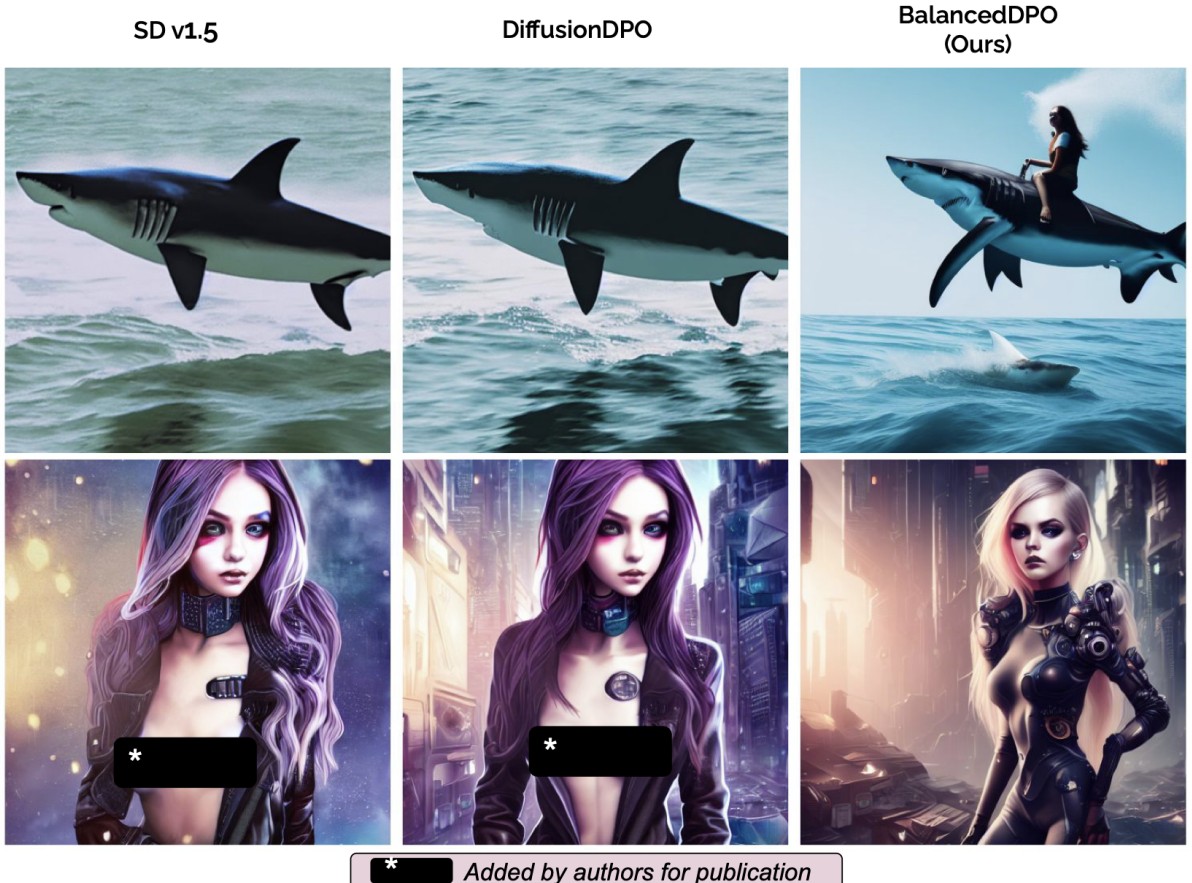

**SD v1.5** | **DiffusionDPO** | **BalancedDPO (Ours)**

\* Added by authors for publication

Figure 7: Comparison of image generations by SD1.5, DiffusionDPO, and BALANCEDDPO (Ours). In the first row (Prompt: "*Person riding a shark*"), BALANCEDDPO accurately includes the person, while SD v1.5 and DiffusionDPO do not. In the second row (Prompt: "*SFW Trustworthy ...*"), BALANCEDDPO generates a high-quality, safe-for-work image, while SD1.5 and DiffusionDPO produce NSFW content. All images are generated with the same seed.

model overfits to a narrow signal. However, in our multi-metric setting, the dynamic update acts as a *moving anchor*. This prevents the model from being pulled excessively toward one specific metric (e.g., maximizing Aesthetic scores while sacrificing Prompt Adherence). Instead, it allows the model to incrementally discover a "consensus manifold" where the requirements of all reward models are satisfied. This mechanism ensures that the KL-divergence constraint remains informative and anchored to the most recent balanced state of the model, leading to the superior win rates observed in Table 4.

**Qualitative Analysis of Metric Bias:** As shown in Figure 2, individual reward models exhibit distinct "corner-case" biases. **Aesthetics-only** training prioritizes photographic flair (e.g., saturation, bokeh) but often suffers from *semantic drift*, sacrificing prompt details for visual appeal. Conversely, **CLIP/PickScore-only** models ensure semantic fidelity but lack artistic polish, often appearing compositionally flat. While **HPS v2.1** provides a stronger baseline, it remains susceptible to stylistic biases in its training data. By requiring a majority consensus, BALANCEDDPO acts as a *consensus filter*. It retains the visual sophistication of aesthetic metrics while using CLIP and HPS signals to ground the imagery in the prompt. This evidence suggests BALANCEDDPO successfully navigates the Pareto front of these conflicting objectives, distilling a holistic generation from heterogeneous reward signals.

**Adherence to prompts during image generation:** In Figure 7, the comparison of image generations by SD1.5, DiffusionDPO, and BALANCEDDPO (Ours) demonstrates a clear advantage of the proposed multi-objective optimization strategy in adhering to user prompts.

In the first row, where the prompt is "*Person riding a shark*" BALANCEDDPO successfully generates an image that includes both the person and the shark, directly aligning with the prompt's requirements. In contrast, both SD1.5 and DiffusionDPO fail to include the person, suggesting their limitations in prompt adherence. These models either omit critical elements or interpret the prompt too loosely, leading to incomplete or inaccurate representations of the requested scene. Similarly, *"SFW Trustworthy ..."*, further highlights the strengths of BALANCEDDPO. While SD1.5 and DiffusionDPO generate NSFW content, BALANCEDDPO reliably produces a safe-for-work image that meets both the prompt's requirements and the expectation for content safety. Overall, these results emphasize the advantage of BALANCEDDPO 's multi-objective optimization strategy, which not only ensures higher fidelity to the specified prompt but also improves prompt adherence.

## 5   Conclusions

In this work, we introduced BALANCEDDPO, a method for aligning text-to-image models with human preferences by optimizing multiple metrics simultaneously. Our experiments show that BALANCEDDPO outperforms existing approaches like DiffusionDPO and SD1.5 across key metrics, including Human Preference Score (HPS), CLIP, PickScore, and Aesthetic quality. While single-metric training leads to imbalances, BALANCEDDPO's multimetric approach ensures balanced, high-quality image generation. The frequent updates to the reference model ($p_{ref}$) further enhance its ability to align with human preferences.

Furthermore, while our evaluation focuses on text-to-image synthesis, the underlying consensus-based mechanism of BALANCEDDPO is inherently modality-agnostic. The strategy of distilling multiple conflicting reward signals into a unified preference signal could be extended to other generative tasks, such as Large Language Model (LLM) alignment for helpfulness and safety, or video generation where temporal consistency and aesthetic quality must be balanced. This suggests that majority-vote preference aggregation is a robust, general-purpose framework for multi-objective alignment.

Overall, BALANCEDDPO offers a scalable and effective solution for multi-metric optimization, demonstrating strong performance across both in-distribution and out-of-distribution datasets, with broad potential for real-world applications.

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

This appendix provides extended insights and results to support the claims made in the main paper.

- In Section A, we describe the datasets used in our experiments, including PartiPrompts, Pick-a-Pic, and HPD, and showcase images generated from randomly sampled prompts to highlight the diversity and quality of the dataset.

- Section B provides more information on Diffusion Model Formulation.

- Section C expands on key findings:
  - Demonstrates the limitations of Vanilla Aggregation for combining metrics, emphasizing the necessity of our proposed BALANCEDDPO method.
  - Presents additional qualitative results on the HPD dataset, showing BALANCEDDPO's ability to generate semantically aligned and visually appealing images compared to SD1.5 and DiffusionDPO.
  - Analyzes the effect of seed variation, highlighting BALANCEDDPO's robustness across random seeds while maintaining semantic alignment and aesthetic quality.

- In Section D, we detail the evaluation procedures used to assess model performance across metrics, ensuring fair and comprehensive comparisons between methods.

## A    Datasets

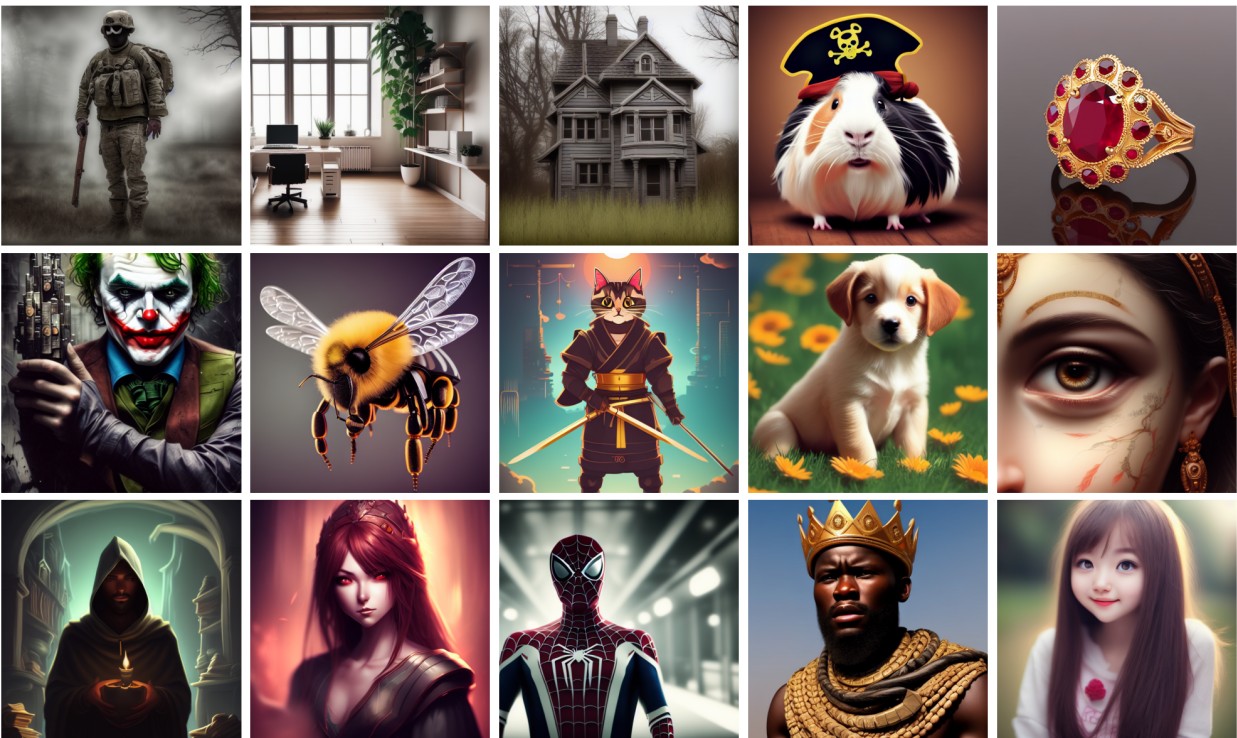

Figure 8: Pick-a-pic (Kirstain et al., 2023) dataset.

For validation, we employ three distinct datasets to provide a comprehensive evaluation:

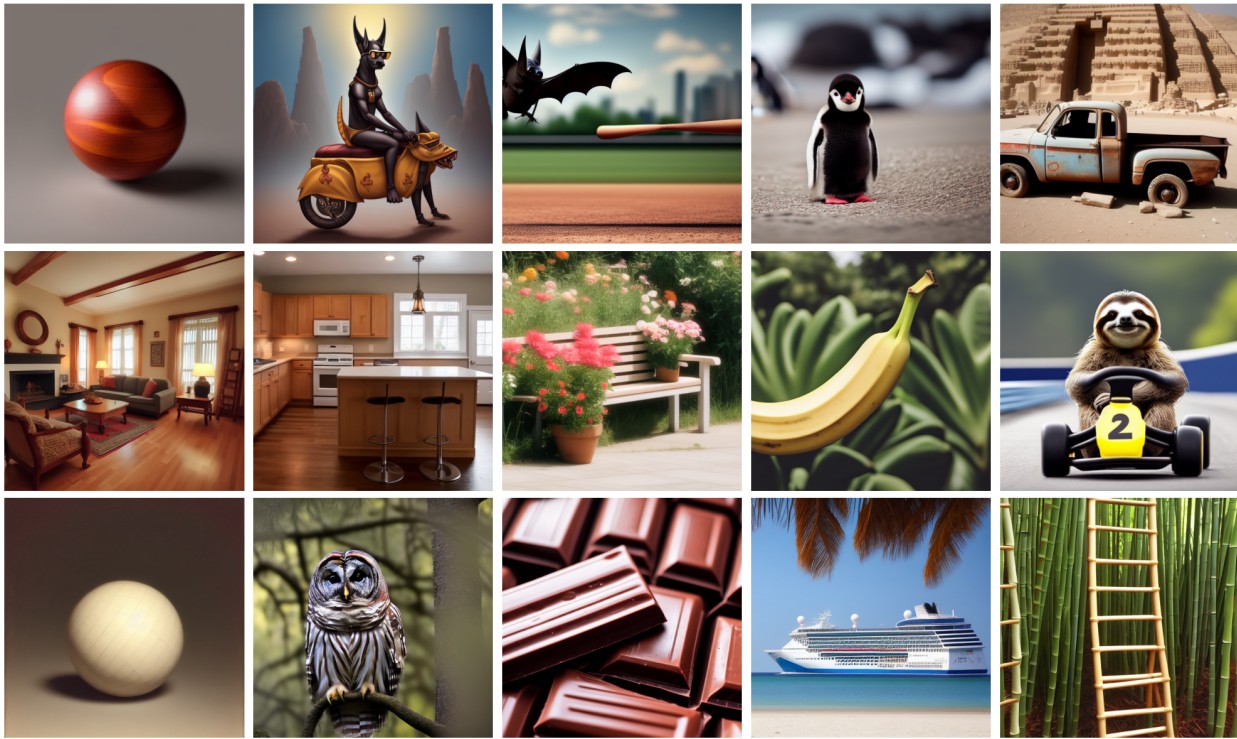

Figure 9: PartiPrompt dataset (Yu et al., 2022)

**Pick-a-Pic**  (Kirstain et al., 2023): The Pick-a-Pic dataset is an extensive, publicly accessible collection of over 500,000 examples, encompassing more than 35,000 unique prompts, each paired with two AI-generated images and corresponding user preference labels. This dataset was compiled through the Pick-a-Pic web application, which enables users to generate images from text prompts and indicate their preferences between image pairs. The primary objective of Pick-a-Pic is to facilitate research in aligning text-to-image generation models with human preferences, thereby enhancing the quality and relevance of generated images. In addition to its use in training, we utilize the validation set for the evaluation, which contains over 500 unique prompts.

**PartiPrompts**  (Yu et al., 2022): PartiPrompts is a comprehensive dataset comprising over 1,600 English prompts, designed to evaluate and enhance the capabilities of text-to-image generation models. Each prompt is categorized across 12 distinct themes—such as abstract concepts, animals, vehicles, and world knowledge—and is further classified into one of 11 challenge aspects, including basic, quantity, words and symbols, linguistics, and complex scenarios.

**Human Preference Dataset v2 (HPD v2)**  (Wu et al., 2023): This dataset serves as the foundation for creating the Human Preference Score v2 (HPS v2), and contains 798,090 human preference choices on 433,760 pairs of images. It is specifically curated to minimize potential biases present in previous datasets and covers a wide range of image sources and styles. It has 400 unique prompts.

We show examples of generated images from prompts sourced from Pick-a-Pic (Figure 8), PartiPrompts (Figure 9) and HPD v2 (Figure 10) below.

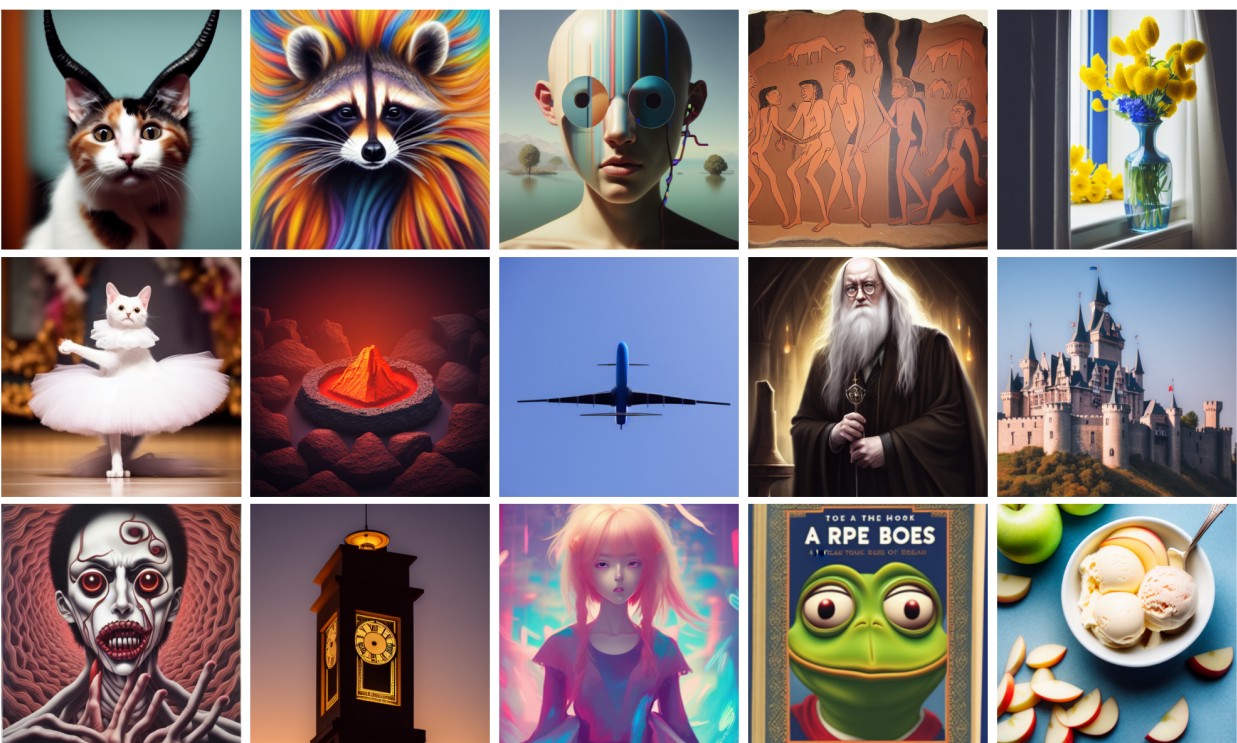

Figure 10: HPD Dataset (Wu et al., 2023)

## B    Diffusion Model Formulation

A generative model class that learns to denoise $\epsilon \sim \mathcal{N}(0, I)$, into structured data. The core component is a denoiser $D$, predicting $\mathbb{E}[x_0|x_t, t]$ given a noisy image $x_t$ at time $t$. The noisy image $x_t$ is as $x_t = \alpha_t x_0 + \sigma_t \epsilon$, where $\alpha_t$ and $\sigma_t$ control noice addition.

Given $q(x_0)$ and noise scheduling functions $\alpha_t$ and $\sigma_t$, the generative model $p_\theta(x_0)$ follows a discrete-time reverse process $p_\theta(x_{0:T}) = \prod_{t=1}^{T} p_\theta(x_{t-1}|x_t)$, with transition dynamics as,

$$p_\theta(x_{t-1}|x_t) = \mathcal{N}(x_{t-1}; \mu_\theta(x_t), \sigma_{t|t-1}^2 I), \tag{12}$$

with $\mu_\theta(x_t)$ being the predicted mean and $\sigma_{t|t-1}^2$ representing the variance at each timestep. This reverse process iteratively refines noisy samples into clean data. The training of diffusion models is performed by minimizing the Evidence Lower Bound (ELBO), which results in objective

$$L_{\text{DM}} = \mathbb{E}_{x_0, \epsilon, t, x_t} \left[ \omega(\lambda_t) \| \epsilon - \epsilon_\theta(x_t, t) \|_2^2 \right], \tag{13}$$

where, $\epsilon \sim \mathcal{N}(0, I)$ represents the noise added to the data, $t$ is sampled uniformly from $[0, T]$, $\epsilon_\theta$ denotes the noise predicted by the model being trained, and $x_t$ is sampled from the distribution $q(x_t|x_0) = \mathcal{N}(x_t; \alpha_t x_0, \sigma_t^2 I)$. The term $\lambda_t = \frac{\alpha_t^2}{\sigma_t^2}$ represents the signal-to-noise ratio at time step $t$, and $\omega(\lambda_t)$ is a pre-specified weight function that adjusts the importance of different timesteps during training.

## C    Extended Results

This section provides additional experiments to support our claims in the main paper. First, we conduct an ablation study to demonstrate the limitations of the Vanilla Aggregation approach, highlighting why

our proposed BALANCEDDPO method is essential for effectively optimizing multiple metrics simultaneously. Second, we present qualitative and quantitative (see Table 1) results on the HPD dataset (Wu et al., 2023), showcasing BALANCEDDPO's ability to generate images that are both semantically aligned with prompts and aesthetically superior compared to SD1.5 and DiffusionDPO. Finally, we analyze the effect of seed variation on image generation, demonstrating that BALANCEDDPO consistently produces high-quality and semantically aligned outputs across diverse prompts, proving its robustness to random initialization.

## C.1 Results on SD 2.1

To further validate the generalization ability of our approach, we evaluate BALANCEDDPO on the Stable Diffusion 2.1 backbone using the Pick-a-Pic dataset (Kirstain et al., 2023). As shown in Table 5, BALANCED-DPO consistently improves over both the base SD 2.1 model and DiffusionDPO across all evaluation metrics, confirming that our multimetric consensus strategy and dynamic reference model updates remain effective on newer diffusion architectures.

Table 5: Comparison of BALANCEDDPO against SD 2.1 and DiffusionDPO on the Pick-a-Pic dataset across four evaluation metrics: HPS, CLIP, PickScore, and Aesthetic.

| Comparison | HPS | CLIP | PickScore | Aesthetic |
|---|---|---|---|---|
| DiffusionDPO vs SD 2.1 | 57.33 | 47.67 | 50.67 | 45.67 |
| BALANCEDDPO vs SD 2.1 | 62.33 | 65.33 | 69.33 | 65.33 |
| BALANCEDDPO vs DiffusionDPO | 53.67 | 65.67 | 69.67 | 70.00 |

## C.2 Generalization to Transformer-based Architectures

To address concerns regarding the modernity of diffusion backbones, we evaluate BALANCEDDPO on **SD3-Medium** (Esser et al., 2024) (using a Multimodal Diffusion Transformer, MMDiT). Since BALANCED-DPO operates at the objective function level by aggregating rewards into a unified consensus signal, it is fundamentally architecture-agnostic.

As shown in Tables 6, BALANCEDDPO provides consistent improvements over the base transformer models and the single-metric DiffusionDPO baseline across all evaluation criteria. Notably, while standard DPO on transformer architectures can sometimes lead to a decline in Aesthetic scores while improving prompt adherence (CLIP), BALANCEDDPO maintains a high win rate across both dimensions. These results demonstrate that the consensus-based alignment strategy generalizes effectively to the latest generation of transformer-based diffusion models.

Table 6: Win rates (%) on SD3-Medium (Transformer backbone) using 500 captions from PartiPrompts.

| Comparison | HPS | CLIP | PickScore | Aesthetic |
|---|---|---|---|---|
| DiffusionDPO vs. Base SD3 | 55.42 | 52.11 | 49.87 | 44.34 |
| BALANCEDDPO vs. Base SD3 | **62.21** | **63.80** | **61.58** | **58.12** |
| BALANCEDDPO vs. DiffusionDPO | **58.45** | **62.27** | **60.11** | **58.35** |

## C.3 Limitations of Vanilla Aggregation

In addition to our proposed multimetric consensus strategy, we experimented with a straightforward approach to combine scores, referred to as Vanilla Aggregation, where individual metric scores (HPS, CLIP, PickScore, Aesthetic) are directly aggregated using a simple weighted sum. As shown in Table 3, this naive approach fails to achieve competitive performance compared to BALANCEDDPO. While Vanilla Aggregation performs slightly better than models trained on individual metrics alone, it struggles to balance competing objectives effectively, resulting in suboptimal performance across multiple metrics.

For instance, Vanilla Aggregation achieves a win rate of 69.70% in HPS and 53.54% in both PickScore and Aesthetic against SD1.5, which is significantly lower than the corresponding scores achieved by BALANCED-

DPO (85.19%, 64.31%, and 69.36%, respectively). Similarly, against DiffusionDPO, Vanilla Aggregation underperforms across all metrics, with a CLIP score of only 26.60% compared to BALANCEDDPO's 73.40%. This performance gap highlights the inability of Vanilla Aggregation to handle the inherent scale differences and conflicts between metrics, leading to suboptimal alignment.

The limitations of Vanilla Aggregation can be further understood through the following optimization equation as shown in Eq. 10:

$$\mathcal{L}_{\text{da}}(\theta) \tag{14}$$
$$= -\mathbb{E}_{(c,x_0^1,x_0^2)\sim\mathcal{D}} \left[ \sum_k w_k \log \sigma \left( \beta \cdot s_k \cdot l_\theta(c, c, x_0^1, x_0^2) \right) \right],$$

where $w_k$ represents the weights for each metric and $s_k$ indicates the preference for the $k^{\text{th}}$ attribute. Similarly, the gradient of the objective boils down to

$$\nabla_\theta \mathcal{L}_{\text{da}}(\theta) \tag{15}$$
$$= -\mathbb{E}_{\mathcal{D}} \left[ \sum_k w_k \beta s_k \sigma \left( -\beta l_\theta(c, x_0^1, x_0^2) \right) \nabla_\theta l_\theta(c, x_0^1, x_0^2) \right].$$

This formulation suffers from issues such as unknown optimal weights ($w_k$), scale dominance by certain metrics, and conflicting gradients that hinder convergence. These challenges make Vanilla Aggregation unsuitable for effectively optimizing multiple metrics simultaneously.

In contrast, BALANCEDDPO's multimetric consensus approach dynamically aggregates preferences through majority voting, ensuring balanced optimization across all metrics without being dominated by any single score. This enables BALANCEDDPO to consistently outperform both Vanilla Aggregation and single-metric models, demonstrating the importance of our proposed method for achieving superior overall performance.

### C.4    Effect of Seed Variation

Figure 11 demonstrates the robustness of BALANCEDDPO across different prompts and seeds, showcasing its ability to generate high-quality and semantically aligned images consistently.

For prompts such as "the Taj Mahal at sunrise" (Parti dataset), BALANCEDDPO produces photorealistic images with varying lighting and perspectives, maintaining semantic alignment across all seeds. Similarly, for "A hybrid creature concept painting of a zebra-striped unicorn with bunny ears and a colorful mane" (HPD dataset), BALANCEDDPO consistently captures the intricate details of the prompt, such as the vibrant colors and unique features of the creature. Prompts like "a calm and peaceful office" (Pick-a-Pic dataset) highlight BALANCEDDPO's ability to render realistic indoor scenes with diverse layouts while adhering to the described atmosphere.

This figure underscores BALANCEDDPO's capability to handle diverse prompts from multiple datasets while maintaining consistency across seeds. The results demonstrate that BALANCEDDPO not only excels in semantic alignment but also generates aesthetically pleasing images with significant variation across seeds, making it a robust solution for text-to-image generation tasks.

### C.5    Scoring Functions

Table 7 presents the average scores for three models - SD1.5, DiffusionDPO, and our proposed BALANCEDDPO-across four key metrics: Human Preference Score (HPS), CLIP score, Pickscore, and Aesthetic score. The results demonstrate that BALANCEDDPO consistently outperforms both SD1.5 and DiffusionDPO across all evaluated metrics. Specifically, BALANCEDDPO achieves the highest average scores in HPS (0.2745), CLIP (0.3771), Pickscore (0.1899), and Aesthetic (4.5034). While the improvements may appear incremental, they represent significant advancements in model performance, particularly given the challenging nature of

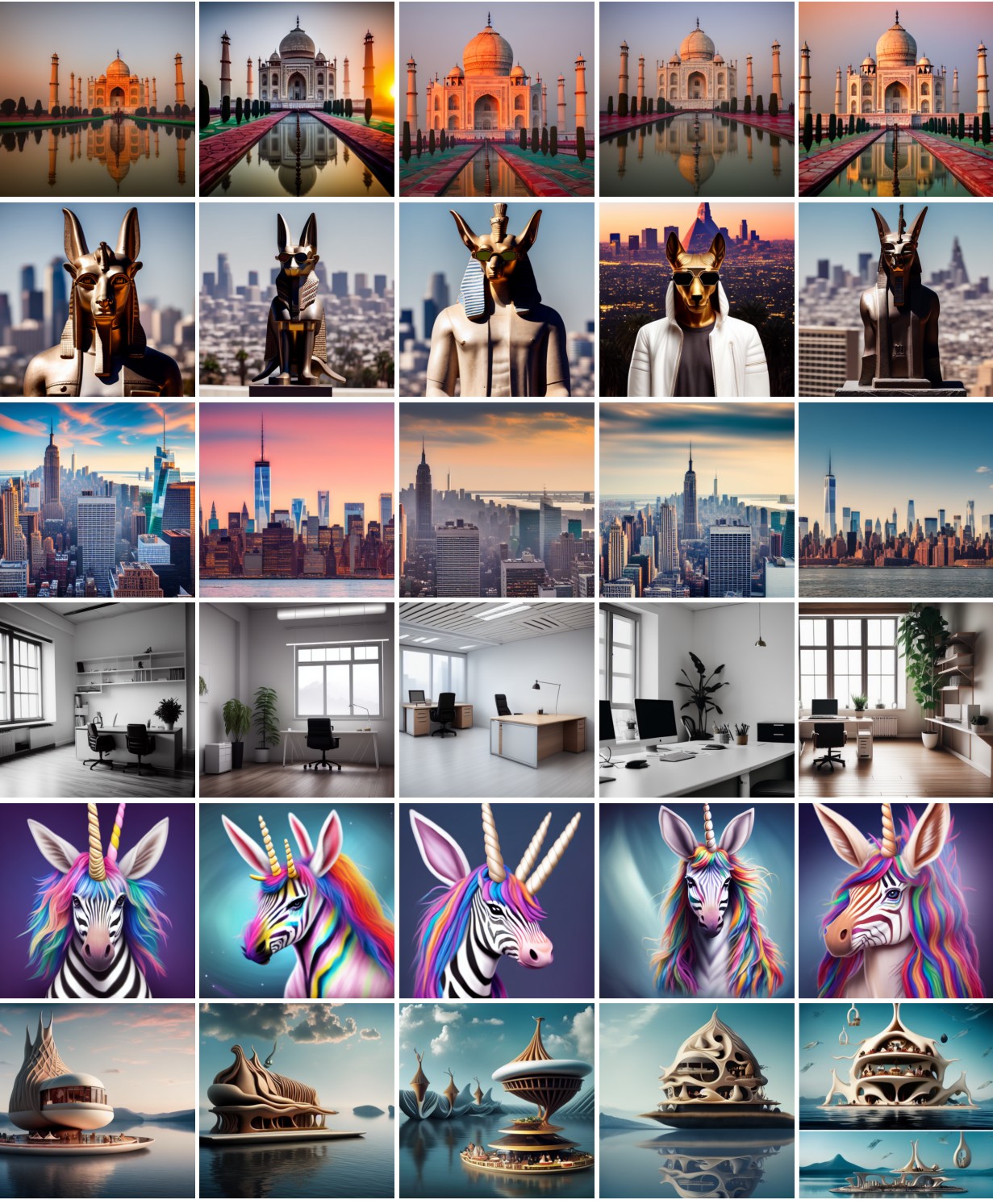

Figure 11: Comparison of images generated using BALANCEDDPO across five different seeds for each prompt. Rows correspond to prompts from the PartiPrompts, Pick-a-Pic, and HPD datasets, while columns represent outputs for different seeds. The figure highlights BALANCEDDPO's ability to consistently generate visually appealing and semantically aligned images across diverse prompts and seeds.

simultaneously optimizing multiple metrics. DiffusionDPO shows slight improvements over SD1.5 in HPS and CLIP scores but performs marginally worse in Aesthetic score. These results highlight the effectiveness of BALANCEDDPO in achieving a balanced optimization across diverse image quality metrics, demonstrating its superior capability in generating images that align well with human preferences while maintaining high visual quality.

Table 7: Average scores for SD1.5, DiffusionDPO, and BALANCEDDPO (Ours) across four key metrics: Human Preference Score (HPS), CLIP score, Pickscore, and Aesthetic score. BALANCEDDPO consistently outperforms the other models across all metrics.

| Model | HPS | CLIP | Pickscore | Aesthetic |
|---|---|---|---|---|
| SD1.5 | 0.2708 | 0.3686 | 0.1897 | 4.4860 |
| DiffusionDPO | 0.2730 | 0.3715 | 0.1898 | 4.4840 |
| BALANCEDDPO (Ours) | **0.2745** | **0.3771** | **0.1899** | **4.5034** |

Table 8 presents statistical measures (mean, minimum, maximum, and standard deviation) for four scoring metrics applied to images generated by the SD1.5 model. These metrics include the Human Preference Score (HPS), CLIP score, Pickscore, and Aesthetic score. The data shows variation in scores across the different metrics, with Aesthetic scores having a notably different scale than the other three metrics.

Table 8: Statistical summary of image quality metrics for SD1.5-generated images across four evaluation scores: Human Preference Score (HPS), CLIP, Pickscore, and Aesthetic.

| Statistic | HPS | CLIP | Pickscore | Aesthetic |
|---|---|---|---|---|
| Mean | 0.2708 | 0.3686 | 0.1897 | 4.4860 |
| Min | 0.2233 | 0.2383 | 0.1656 | 4.0969 |
| Max | 0.3114 | 0.5161 | 0.2195 | 4.7878 |
| Std | 0.0130 | 0.0461 | 0.0086 | 0.1049 |

## D    Evaluation Procedure

We evaluate BALANCEDDPO in comparison to Stable Diffusion 1.5 (SD1.5) (Rombach et al., 2022) and DiffusionDPO (Wallace et al., 2024), addressing the limitation of these approaches in simultaneously improving scores across all metrics.

1. **Image Generation:** For each caption from our validation datasets, we generate five images using different random seeds for all models (BALANCEDDPO, SD1.5, and DiffusionDPO).

2. **Metrics Used:** We evaluate generated images using four key metrics: Human Preference Score (HPS), CLIP score, PickScore, and Aesthetics score following DiffusionDPO.

3. **Best Score Selection:** For each metric and prompt, we select the highest score from the five generated images for all the models. This method mitigates random variations in image generation, ensures each model demonstrates its best performance, and maintains consistency across all models by automating the selection process.

4. **Win Rate Calculation:** We computed the win rate for each model pair comparison (BALANCEDDPO *vs.* SD1.5, BALANCEDDPO *vs.* DiffusionDPO, and DiffusionDPO *vs.* SD1.5). The win rate represents the proportion of times a model's best score is preferred (*i.e.*, higher) compared to the other model's best score for the same prompt and metric.

## Ethics

While our model, BALANCEDDPO, demonstrates significant advancements in text-to-image generation, it is crucial to acknowledge the ethical considerations associated with its use. Text-to-image models, including

ours, inherently carry risks such as potentially generating harmful, biased, or inappropriate content. These risks may arise from biases present in the training data or unintended misuse of the model. We emphasize that it is the responsibility of users to employ this technology ethically and avoid generating content that is harmful, hateful, or otherwise inappropriate. Our model is designed for research and creative purposes, and we strongly discourage its use for generating misleading, offensive, or exploitative content.

