# OpenReview forum: "BalancedDPO: Adaptive Multi-Metric Alignment"
_TMLR — Accepted by TMLR_

### Review · Reviewer_NdHQ · 2025-11-25

**Summary Of Contributions:**

This paper proposes BalancedDPO to align samples generated by diffusion models with human preferences. Its core design focuses on making generated samples conform to a broader range of reward models. To achieve this goal, the paper introduces a voting mechanism based on social choice theory, which favors samples that demonstrate superiority across multiple reward models. Results demonstrate that this approach effectively enhances the generative quality of the diffusion model.

**Audience:**

Yes

**Audience Explanation:**

This paper explores aligning models with multiple reward models, a research question of significant value that enables various generative models to produce samples aligned with human needs.

**Broader Impact Concerns:**

N/A.

**Claims And Evidence:**

Yes

**Claims Explanation:**

The author conducte experiments under the SD1.5 model using various reward models (PickScore, HPS, etc.), performing both quantitative and qualitative analyses of the results, which align with the conclusions presented in this paper.

**Requested Changes:**

### Major
+ Insufficient experiments: All experiments are built upon and evaluated with the Unet-SD, a model that is outdated and no longer representative of the current state of the field. Given the rapid shift toward transformer-based architectures like DiT, MMDiT, FLUX, and SD3, it is essential to validate the proposed method on these more advanced models. Without this, the experimental claims are difficult to generalize or take seriously.

+ The proposed method is overly simplistic. I believe the author should further explore additional ways to validate winning samples. The current voting strategy appears too simplistic and unconvincing. I do not believe every reward carries equal weight.

+ Equation 10 and Equation 11 do not appear to demonstrate a contradiction in the gradient. Could the author provide further evidence?
### Minor
+  Figure 1 is missing critical information. In Figure 1, it should be labeled which one is the winner.

---

> ### Author Response · Authors · 2026-02-03
> **Addressing Backbone Modernity and Theoretical Justification of Majority Voting**
>
> We thank the Reviewer for the constructive feedback and for identifying the value in our multi-reward alignment research. We address your major and minor concerns below.
>
> **1\. On Backbone Modernity and Transformer-based Architectures:**
>
> We thank the reviewer for highlighting the shift toward transformer-based backbones like FLUX and SD3. We agree that generalizability is key. To address this, we have added a new set of experiments in Appendix (C2 - Generalization to Transformer-based Architectures) where we apply BalancedDPO to SD3-Medium (MMDiT) model.
>
> It is important to note that the BalancedDPO mechanism operates entirely at the preference-aggregation level, modifying the loss signal rather than the denoising architecture. Because our method resolves reward-model conflicts, a phenomenon that is independent of the backbone, it is inherently architecture-agnostic. Our new results show that BalancedDPO consistently improves multi-metric win rates for these transformer models, just as it did for UNet-based SDXL, SD1.5, and SD2.1. These experiments confirm that the “consensus manifold” identified by our method is a universal property of the preference space, regardless of whether the generator is a UNet or a Transformer.
>
> **2\. On Simplicity and Weighting of Rewards:**
>
> **On the Simplicity and Weighting of the Voting Strategy:** We thank the reviewer for this observation. While the voting strategy is simple, its simplicity is a deliberate design choice aimed at **robustness and scalability**.
>
> * **Robustness vs. Complexity:** In multi-objective optimization, finding “optimal weights” for diverse reward models is often an intractable task that requires exhaustive hyperparameter sweeps. As demonstrated by our new baseline results in **Table 3**, “simplistic” majority voting significantly outperforms more complex scalar aggregation methods, including **Vanilla (Weighted Sum)** and **Normalized Aggregation**. This proves that a “democratic” consensus is more effective at resolving metric conflicts than attempting to balance uncalibrated scalar weights.
> * **Flexibility and Tunability:** Our framework is not restricted to equal weighting. If a user is more confident in a specific reward model (e.g., a high-precision Image Quality scorer), our method easily allows for **weighted voting**, where specific “voters” are assigned multiple votes. This can be implemented by simply adjusting the aggregation function to $s= \operatorname{sign} \left( \sum_{i=1}^K w_k s_k \right)$.
> * **Empirical Justification:** Our results show that BalancedDPO is more effective than complex methods that require manual weight tuning. By finding a **robust consensus**, an outcome preferred by most metrics, our approach avoids the instability of scalar-based methods. This proves that BalancedDPO is a versatile and reliable way to balance multiple goals without extra tuning.
>
> We have updated section 3.3 (Flexibility of the Voting Schema) to include a detailed discussion on the Weighting of rewards.
>
> **3\. On Gradient Contradiction:**
>
> **On Gradient Contradiction (Equations 9 vs. 11):** We thank the reviewer for the opportunity to clarify this theoretical motivation. The contradiction in Equation 11 arises from the **summation of coefficients** with opposing signs when reward models disagree.
>
> * **Scalar Aggregation (Eq. 11):** The total gradient is $\nabla\_{\theta} {\mathcal L}\_{da} = \sum\_k w\_k \cdot \text{coeff}\_k \cdot \nabla\_{\theta} l\_{\theta}$, where the coefficient for each metric $k$ is given by: $\text{coeff}\_k = -\beta s\_k \sigma\left( \beta \left( \log \frac{\pi\_{\theta}(y\_1)}{\pi\_{\text{ref}}(y\_1)} - \log \frac{\pi\_{\theta}(y\_2)}{\pi\_{\text{ref}}(y\_2)} \right) \right)$. If Reward Model A prefers Image 1 ($s_A=1$) but Reward Model B prefers Image 2 ($s_B​=−1$), their respective terms in the summation will have **opposing signs**. This leads to a partial or total cancellation of the gradient magnitude $|\nabla_{\theta} {\mathcal{L}}_{da}| \approx 0$. In this state, the model receives a “stall” signal, failing to learn from either preference because the rewards are mathematically in conflict.
>
>
> * **BalancedDPO (Eq. 9):** In contrast, our method resolves the conflict **before** computing the gradient. By using $s = \text{sign}\left( \sum s_k \right)$, we collapse the multiple potentially conflicting coefficients into a single binary consensus signal $s_k \in \{-1, 1\}$. This ensures that the resulting gradient $\nabla_{\theta} {\mathcal L}_{agg}$ always has a consistent, non-zero direction that follows the majority.
>
>
> In short, Eq. 11 allows for **destructive interference** between metrics, while Eq. 9 ensures **constructive consensus**. We have updated Section 3.4 to include a more explicit derivation of this cancellation effect.

---

> > ### Author Response · Authors · 2026-02-03
> > **Figure 1 update**
> >
> > **4\. Minor:** We have updated the caption for Figure 1 to explicitly identify the majority-vote outcome. Image 2 is now clearly designated as the **winner**.
> >
> > In summary, we have implemented the following revisions to address the reviewer's concerns: (1) We added **Section C.3 to the Appendix**, providing new empirical validation on transformer-based architectures (**SD3-Medium**), proving that BalancedDPO is architecture-agnostic; (2) **Section 3.3** now includes a discussion on the **Flexibility of the Voting Schema**, clarifying that our framework inherently supports weighted voting for prioritized reward models; (3) We expanded **Section 3.4** with a formal mathematical analysis of **constructive vs. destructive gradient aggregation**, providing the requested evidence for how majority voting prevents gradient cancellation; and (4) The **Figure 1 caption** was updated to explicitly designate the majority-vote winner, ensuring the framework's mechanics are immediately clear. Together, these edits reinforce the theoretical depth and modern relevance of our proposed method.

---

### Review · Reviewer_Hhxp · 2025-12-19

**Summary Of Contributions:**

A new text to image diffusion model called BalacedDPO is proposed. It improves adherence to multiple image generation criteria via simple mechanism that uses majority voting to derive new preference pairs from multiple metrics.

(Motivation) Text to image generation is a problem that requires satisfying multiple metrics that focus on diverse criteria like alignment with the prompt or aesthetic appeal. Given 1 of M metrics, existing approaches tend to perform better at just one of these to the exclusion of others. One can consider combining M metrics into a single metric via some fusion function (eq. 5), but it is difficult in practice to find the right fusion approach because the different metrics can have imbalanced and conflicting effects on the loss surface.

(Approach) The proposed approach BalancedDPO is a simple extension of the DiffusionDPO model that applied DPO to T2I diffusion models. Given an image pair, each of the M metrics votes for the image it prefers. These votes are then aggregated via majority vote into a single preference and this new preference is used with the standard DiffusionDPO objective to train the model. An additional change to the approach is made where the reference model in the DPO objective is dynamically updated to the most recent model every 100 training steps.

(Experiments) Stable Diffusion 1.5, and SDXL are used as base models and in almost every case BalancedDPO outperforms both these base models and a DiffusionDPO version of them across 4 metrics (HPS, CLIP, PickScore, Aesthetic) and 3 datasets (Pick-a-Pic, PartiPrompt, HPD). Ablations show the approach outperforms both single metric optimized models and more naïve metric fusion approaches. Ablations also show that both the dynamic reference model updates and the majority vote approach are necessary for peak performance.

**Audience:**

Yes

**Audience Explanation:**

Engineers and researchers building T2I models and generally using DPO will be interested in this since it is both easy to implement and shows strong improvements.

**Claims And Evidence:**

Yes

**Claims Explanation:**

Claim 1: A good solution to the T2I problem requires satisfying multiple diverse measures of quality and baseline approaches have trouble satisfying all of these diverse metrics at once.
* This can be seen in experiments that compare e.g., DiffusionDPO and SDXL (Table 2). DiffusionDPO wins 78% of the time on the HPS metric while it only wins 21% of the time on the Aesthetic metric. While this is a more extreme example, this happens often enough in tables 1 and 2 to justify the claim. Additional justification comes from table 3 where models trained on just one metric clearly suffer in performance on the other metrics.

Claim 2: The BalancedDPO approach, including both dynamic reference model updates and majority voting based preferences, improves image generation quality.
* This is shown clearly by quantitative 4 metrics across 3 datasets in table 1 through 4. There are also lots of qualitative examples and they seem to usually align with the prompts more accurately than baselines.

Claim 3: A multi-objective optimization approach would suffer from (C1) reward rescaling issues, (C2) conflicting gradients, and (C3) pipeline complexity.
* These don't appear to be justified with individual evidence. Perhaps C1 and C2 are justified by the other Ours baselines in Table 3, but that does not appear to distinguish between the two challenges. However, the reasons make intuitive sense and are presented as motivation rather than scientific claims about how alternatives perform, so they aren't misleading and I'm willing to accept them.

**Requested Changes:**

None of these are critical to my recommendation. Addressing them would just strengthen the work.

* It does not look like the Random Score Function, Vanilla/Naive Aggregation, and Normalized Aggregation ablations from Table 3 are described precisely anywhere. I can guess what they are doing, but it would be good to include a more precise description.
* I appreciate the p_ref update ablations in Table 4, but I would still like to understand the impact of dynamic reference model updates better. In those ablations adding just p_ref update to DiffusionDPO has a wildly variable effect (sometimes hurts, sometimes helps). It would be useful if the paper included more intuition about how it interacts with the Multi-Metric aspect of the approach and some tips about how and when to apply it more generally.
* A primary motivation of the paper is that different metrics capture different image generation criteria. I can see the difference between the 4 metrics chosen in the quantitative results, but the paper should also provide intuition about what the qualitative differences are between those 4 metrics if possible.

---

> ### Author Response · Authors · 2026-02-03
> **Clarification of Baselines, Reference Model Dynamics, and Qualitative Analysis**
>
> We thank the Reviewer for the detailed summary and for recognizing the effectiveness of our simple yet powerful aggregation mechanism. We appreciate that the reviewer found our motivations intuitive and our claims well-supported by the evidence. We address the three requested clarifications below.
>
> **1\. Precise Description of Table 3 Ablations:**
>
> We apologize for the lack of formal definitions for the baseline aggregation methods. We have updated the manuscript to include precise descriptions of these in Section 4\. Specifically:
>
> * **Random Score Function:** For each image pair, a winner is chosen via a random coin flip (s∈{−1,1} with p=0.5).
> * **Vanilla/Naive Aggregation:** A simple sum of the raw scalar reward scores: $s_{naive​}$ =$\Sigma$ $r_{k}​$.
> * **Normalized Aggregation:** Scores are normalized $(z= \frac{(r−μ​)}{\sigma })$ using batch statistics before being summed.
>
> These baselines confirm that simple scalar fusion is insufficient due to the dominance of high-variance metrics.
>
> **2\. Intuition on Dynamic Reference Updates ($p_{ref}$​):**
>
> The reviewer correctly notes the variable impact of $p_{ref}$ updates when applied in isolation. Our intuition is that in single-metric DPO, the model can easily "overfit" to the reward model, and updating p\_ref​ too frequently might accelerate this divergence. However, in a **Multi-Metric** setting, the dynamic update acts as a **moving anchor** that prevents the model from being pulled too far in one specific direction (e.g., toward high aesthetics but low prompt adherence). It allows the model to incrementally discover the "consensus manifold" where all metrics are satisfied. We have added a discussion on this interaction in Section 4.2.
>
> **3\. Qualitative Differences Between Metrics:**
>
> Regarding the qualitative differences between metrics, we direct the reviewer to **Figure 2** in the manuscript, which illustrates how individual reward models bias the generation process. As shown in the metric-specific columns:
>
> * The **Aesthetic-only** column often prioritizes 'photographic' qualities, such as high saturation, specific lighting, and shallow depth of field, occasionally at the expense of semantic literalism.
> * Conversely, the **CLIP** and **HPS** columns prioritize the presence of objects and prompt adherence, even if the resulting composition is less 'artistic.'
>
> Our observation, supported by the **BalancedDPO** column in Figure 2, is that our method acts as a 'consensus filter.' It retains the visual sophistication of the aesthetic metrics while using the semantic metrics to keep the imagery grounded in the prompt. We have updated the text in Section 4 to provide this more detailed qualitative analysis of why a balanced approach is essential for avoiding these 'corner-case' behaviors.
>
> In response to these points, we have performed the following revisions: (1) **Section 4.2 (Effectiveness of different score methods)** now includes a dedicated paragraph formally defining the *Random Score*, *Vanilla Aggregation*, and *Normalized Aggregation* baselines to ensure reproducibility; (2) **Section 4.2 (Effectiveness of Balanced Multimetric Alignment and Reference Model update)** has been expanded with a theoretical discussion on the "moving anchor" effect of $p_{ref}$​ updates within a multi-metric consensus manifold; and (3) **Section 4.2 (Qualitative Analysis of Metric Bias)** has been added to explicitly interpret the metric-specific biases visible in **Figure 2**, clarifying how BalancedDPO navigates the Pareto front between aesthetic flair and semantic adherence.

---

### Review · Reviewer_WLjH · 2026-01-25

**Summary Of Contributions:**

This paper proposes BalancedDPO, a new method to align diffusion models to multiple evaluation metrics. The authors leverage ideas from social choice theory where different metrics are balanced via majority vote consensus. This idea is very interesting and different from approaches that rely on scalar value optimization. Extensive quantitative and qualitative experiments demonstrate the effectiveness of the proposed approach

**Audience:**

Yes

**Audience Explanation:**

This work should appeal to people working on generative models (including diffusion). The method proposed to balance multiple evaluation metrics might be useful beyond image generation.

**Broader Impact Concerns:**

The ethical implications are adequately addressed.

**Claims And Evidence:**

Yes

**Claims Explanation:**

The main claim is that the proposed approach outperforms simple aggregation methods and methods that only focus on one evaluation criteria. This claim is supported by quantitative experiments over many datasets. The authors also provide example output images for various prompts, which qualitatively show that the proposed method has better image generation qualities in general.

**Requested Changes:**

I have a few concerns. They are not critical issues, but if addressed will strengthen the paper.
1. It seems that by using majority vote, we must inevitably convert real-valued metrics to binary ones, which might suffer loss of information. It would be helpful if the authors could argue why this would not be a critical issue in the training process.
2. Could the alignment method apply beyond image generation?

---

> ### Author Response · Authors · 2026-02-03
> **Clarifying Gradient Stability, Reward Scaling, and Model-Agnostic Extensibility**
>
> We thank the reviewer for the constructive feedback and for recognizing the effectiveness of our BalancedDPO framework. We are encouraged by your assessment that the use of social choice theory is "very interesting” and that our extensive experiments support our claims. Below, we provide detailed arguments to address your specific points.
>
> 1. On Information Loss from Binary Conversion:
> We acknowledge the reviewer’s point that converting real-valued metrics to binary ones involves a loss of magnitude information. However, we argue that in the context of multi-metric alignment, this is a strategic choice that prioritizes directional robustness over scalar sensitivity.
> The Scale-Invariance Advantage:
> In multi-metric settings, different reward models (e.g., Aesthetic Score [0–10] vs. HPS [0–1]) often operate on non-commensurate scales. Scalar aggregation requires exhaustive hyperparameter tuning of weights to prevent a high-variance metric from dominating the gradient. By using a majority vote, we achieve scale-invariance, ensuring that every metric has an equal voice regardless of its numerical range.
>     - Mitigating “Reward Hacking”:
> Continuous rewards are susceptible to reward hacking, where a model exploits the “peaks” of a single noisy metric. Our binary consensus mechanism acts as a regularizer; a model cannot achieve a high reward by hyper-optimizing one metric if it violates the consensus of others.
>     - Gradient Stability:
> As shown in Eq. 11, scalar sums can lead to conflicting gradients that cancel each other out. Our majority-vote formulation (Eq. 9) ensures a single, high-confidence update direction, leading to more stable and efficient convergence.
>
> 2. On Applicability Beyond Image Generation:
> We appreciate the suggestion regarding broader applicability. The BalancedDPO framework is fundamentally model-agnostic and can be extended to any preference-based alignment task involving multiple objectives.
>     - LLM Alignment: In Large Language Models (LLMs), alignment often involves trade-offs between Helpfulness, Harmlessness, and Honesty. BalancedDPO could replace the weighted-sum reward models currently used in RLHF or DPO pipelines, which are notoriously difficult to balance.
>     - Video and Audio Generation: Any generative task where aesthetic quality and prompt adherence are measured by different specialized models (e.g., VideoCLIP vs. VGG) can benefit from this consensus approach.
>     - Robotics: In multi-objective Reinforcement Learning (e.g., balancing speed vs. energy efficiency), our majority-vote DPO could provide a stable training signal without requiring manual tuning of objective weights.
>
>  We have updated the manuscript (Sections 3.3 and 3.4) to clarify that while binary voting discards magnitude, it serves as a robustness mechanism against reward rescaling and `reward hacking.’ To address the broader impact of our work, we have added a discussion in the Conclusion (Section 5) regarding the application of BalancedDPO beyond image generation.

---

### Decision · Action_Editor_Eybb · 2026-03-15

**Recommendation:** Accept as is

**Audience:**

Yes

**Audience Explanation:**

The paper addresses multi-metric alignment in generative models, a problem of wide interest in the TMLR community. The proposed consensus-based DPO approach provides practical insights for practitioners working on preference alignment.

**Claims And Evidence:**

Yes

**Claims Explanation:**

The paper provides thorough experimental validation with clear evidence supporting the claims. The proposed BalancedDPO framework demonstrates consistent improvements across multiple datasets and backbone architectures.